# Mixture of Attentions for Speculative Decoding

**Matthieu Zimmer**[*]**, Milan Gritta**[*] **& Gerasimos Lampouras**
Huawei Noah's Ark Lab, `firstname.lastname@huawei.com`

**Haitham Bou Ammar**[‡]
Huawei Noah's Ark Lab, UCL Centre for Artificial Intelligence

**Jun Wang**[‡]
UCL Centre for Artificial Intelligence

## Abstract

The growth in the number of parameters of Large Language Models (LLMs) has led to a significant surge in computational requirements, making them challenging and costly to deploy. Speculative decoding (SD) leverages smaller models to efficiently propose future tokens, which are then verified by the LLM in parallel. Small models that utilise activations from the LLM currently achieve the fastest decoding speeds. However, we identify several limitations of SD models including the lack of on-policyness during training and partial observability. To address these shortcomings, we propose a more grounded architecture for small models by introducing a Mixture of Attentions for SD. Our novel architecture can be applied in two scenarios: a conventional single device deployment and a novel client-server deployment where the small model is hosted on a consumer device and the LLM on a server. In a single-device scenario, we demonstrate state-of-the-art speedups improving EAGLE-2 by 9.5% and its acceptance length by 25%. In a client-server setting, our experiments demonstrate: 1) state-of-the-art latencies with minimal calls to the server for different network conditions, and 2) in the event of a complete disconnection, our approach can maintain higher accuracy compared to other SD methods and demonstrates advantages over API calls to LLMs, which would otherwise be unable to continue the generation process.

## 1 Introduction

Auto-regressive inference with LLMs has become quite cost-prohibitive due to the increasing parameter count of recent transformer-based LLMs (Vaswani, 2017). Different types of (usually orthogonal) solutions have been proposed to address this challenge, e.g. Mixture of Experts (Jacobs et al., 1991), Flash Attention (Dao et al., 2022), Model Quantization and Distillation (Polino et al., 2018), Linear/Sparse Self-Attention (Zhang et al., 2021), Tensor Parallelism (Shoeybi et al., 2019) and others. In this work, we focus on a recent LLM acceleration technique called Speculative Decoding, which leverages efficient models (smaller but less capable) to draft future tokens, which are verified by the LLM (more capable but much less efficient) in parallel (Leviathan et al., 2023).

The most recent state-of-the-art SD methods, like EAGLE (Li et al., 2024b) and MEDUSA (Cai et al., 2024a), leverage activations from the LLM. However, those methods have some architectural limitations including partial observability and the lack of on-policyness. Partial observability occurs when the small (draft) model lacks complete information about the state of the LLM, leading to suboptimal predictions. The lack of on-policyness during training arises because the small model is often trained under ideal conditions, assuming perfect inputs. This does not reflect the real-world scenario where the small model generates some inputs. The longer we draft new tokens using only

---

[*]These authors contributed equally to this work
[‡]Corresponding authors: haitham.ammar@huawei.com, jun.wang@cs.ucl.ac.uk

the small model, the bigger the distribution shift from the training setting. These limitations can degrade the performance and reliability of speculative decoding.

To address these challenges, we propose a novel architecture for speculative decoding that enhances the small model's ability to accurately predict future tokens and aligns its training more closely with the inference process. Our architecture introduces several key improvements, including Layer Self-Attention (LSA) to mitigate partial observability, Cross-Attention (CA) to improve on-policyness and training efficiency, and a flexible Target Layer Inference (TLI) mechanism to balance computational efficiency and prediction accuracy. We evaluate our approach in the standard single-device setting where we demonstrate state-of-the-art speedups.

Furthermore, the SD paradigm is also ideal in the following scenario a) the model size is limited by some external factor e.g. the computational capabilities of a client device, and b) we can assume access to a larger model e.g. an LLM hosted on a server. Under this paradigm, the goal is to minimise server-side inference as well as to maintain high accuracy in the event of a total disconnection. This is an important consideration because it could pave a way for serving LLMs on edge devices, enabling them to generate responses offline while leveraging the capabilities of the large model. To this end, we extend our methodology to a client-server scenario. In this setting, we demonstrate state-of-the-art latency and minimal server calls under various network conditions (4G, 5G). Our method maintains a higher accuracy in the event of a disconnection, making it a preferred choice over independent small models or API calls that would be unable to continue generation.

**Contributions** We introduce a Mixture of Attentions architecture for Speculative Decoding that addresses current limitations such as partial observability as well as enabling efficient (more on-policy) training while being auto-regressive. We reuse additional activations from the LLM in the small model, enabling a trade-off between drafting speed and response quality. We conduct extensive experiments to demonstrate the effectiveness of our approach. Compared to EAGLE-2, we show a 9.5% decoding speedup with a 25% higher acceptance rate in a single-device scenario and a 84% speedup with a 53% higher acceptance rate in a client-server scenario. Finally, we propose a new framework for LLM serving in speculative client-server settings and show its effectiveness.

## 2 BACKGROUND

We first present the background knowledge required for the remainder of the paper, i.e. the decoding mechanisms of LLMs as well as the drafting + verification techniques that ensure correct generation.

### 2.1 LLM DECODING

Decoding refers to the process by which LLMs generate tokens in response to input queries. This generation is typically done auto-regressively, where each new token $y_t$ is sampled from the LLM's distribution, conditioned on both the query and the preceding tokens $y_{<t}$. We explore decoding from the perspective of dynamic systems, providing a foundation for developing new decoding mechanisms that combine large and small models (Kong et al., 2024). The internal workings of LLMs can be best understood from a dynamic system perspective, which evolves as tokens are generated. Given a large model $\mathcal{M}_{\text{Large}}$, we can describe the state transition model of vanilla decoding as:

$$\boldsymbol{h}_{\leq t+1}, \boldsymbol{o}_{t+1} = f_{\text{Large}}(\boldsymbol{h}_{\leq t}, token\_embed(y_t)), \qquad y_{t+1} \sim \text{Softmax}(LM\_head(\boldsymbol{o}_{t+1})), \qquad (1)$$

where $\boldsymbol{h}_{\leq t}$ represents the key and value tensors of every layer until the current time-step $t$, $y_t$ is the most recent token and $y_{t+1}$ is the next token, which is sampled from a softmax distribution. Furthermore, *token_embed* is a lookup table, it assigns an embedding to a particular token of the vocabulary $\mathcal{V}$. *LM_head* is a projection from the embedding size to the vocabulary size $|\mathcal{V}|$. Finally, $f_{\text{Large}}(\cdot)$ is the function aggregating all decoder layers of $\mathcal{M}_{\text{Large}}$ and $\boldsymbol{o}_{t+1}$ is the activation of the final decoder layer. With this, the state of the dynamic system is composed of $(\boldsymbol{h}_{\leq t}, y_t)$, the minimal information needed to sample the next token from $\mathcal{M}_{\text{Large}}$.

### 2.2 SPECULATIVE DECODING

Some of the earliest work on speculative decoding was introduced by Stern et al. (2018), later extended to non-greedy sampling (Leviathan et al., 2023). These methods are motivated by the pro-

hibitive cost of auto-regressive generation with $\mathcal{M}_{\text{Large}}$ that could be alleviated by using a draft model $\mathcal{M}_{\text{Small}}$ that can more efficiently generate tokens that do not require the full capability of $\mathcal{M}_{\text{Large}}$. These hypotheses, commonly referred to as drafts, can then be verified in parallel with $\mathcal{M}_{\text{Large}}$ using rejection sampling (Leviathan et al., 2023), i.e. discard all tokens after the first mismatch. We follow the standard draft and verification cycle that iterates over the following two steps until the "end-of-sequence" token or the maximum sequence length has been reached.

1. $\mathcal{M}_{\text{Small}}$ generates new tokens $y_{t+1}, \cdots, y_{t+K}$ where $K$ is the length of the draft sequence, auto-regressively (Xia et al., 2023; Li et al., 2024b) or in parallel (Ankner et al., 2024). The number of tokens $K$ is typically fixed, which was the standard approach until recently when dynamic $K$ was proposed (Nair et al., 2024; Mamou et al., 2024; Huang et al., 2024).

2. Verify $K$ drafted tokens in a single forward pass with $\mathcal{M}_{\text{Large}}$. Verification uses either greedy (exact) matching (Xia et al., 2023), speculative sampling (Zhou et al., 2023; Leviathan et al., 2023) or 'approximate' verification (Stern et al., 2018) which relaxes the acceptance criteria but does not guarantee $\mathcal{M}_{\text{Large}}$'s output distribution. In all cases, after the first rejection, all subsequent/future tokens are typically discarded.

When the drafting of $y_{t+1}, \cdots, y_{t+K}$ is done auto-regressively, token by token, we refer to this strategy as *chain drafting*. Several works (Cai et al., 2024b; Li et al., 2024b) extended this method to *tree drafting* for additional optimisation. In this case, multiple tokens $y_{t+i}$ can be proposed for every future position $i$. Verification is performed with Tree Attention (Miao et al., 2024) to efficiently handle multiple branching paths proposed by tree drafting. Consequently, this leads to an increase in acceptance lengths and reduces the number of calls to $\mathcal{M}_{\text{Large}}$ compared to chain drafting. For small batch sizes, the LLM generation is memory-bound, this is where speculative decoding can better leverage the spare compute especially with Tree Attention.

In this paper, we employ tree drafting from EAGLE-2 (Li et al., 2024b) to construct trees with a variable structure. Starting from the root node, we expand the $B$ most probable tokens from the model $\mathcal{M}_{\text{Small}}$ $(y_{t+1}|\cdot)$. Then, for a fixed depth $D$, we recursively perform the following steps: for each existing branch, we compute the joint probability $\prod_{t \in D} \mathcal{M}_{\text{Small}}(y_t|y_{t-1}, \cdots)$ of their $B$ child tokens, leading to $B^2$ expansions as we have $B$ branches, each with $B$ children. From the $B^2$ expansions, we select the top $B$ branches based on their joint probabilities for the next tree layer expansion. Upon reaching the maximum depth, we retain up to $m$ tokens from the total $B + (D-1)B^2$ nodes, selecting those with the highest joint probability for verification.

## 2.3 ARCHITECTURE OF $\mathcal{M}_{\text{SMALL}}$

Speculative decoding architectures broadly fall into two categories, *independent* and *self-drafting*. Independent drafters are typically smaller versions of $\mathcal{M}_{\text{Large}}$ from the same model family (Li et al., 2024a; Zhao et al., 2024; He et al., 2023) while self-drafting methods leverage either a subset of $\mathcal{M}_{\text{Large}}$ and/or newly initialised parameters (Ankner et al., 2024; Cai et al., 2024a).

Our contribution is built on EAGLE (Li et al., 2024b), a *self-drafting* architecture which has shown the best results on the Spec-Bench (Xia et al., 2024) leaderboard so far. The drafter reuses the *token_embed* and *LM_head* parameters of $\mathcal{M}_{\text{Large}}$ (1). It takes as input the ground-truth activations of the last decoder layer of $\mathcal{M}_{\text{Large}}$, $\boldsymbol{o}_1, \cdots, \boldsymbol{o}_t$ and the tokens of the sequence $y_1, \cdots, y_t$ to predict the next activations $\hat{\boldsymbol{o}}_{t+1}$, which is passed to the *LM_head* to predict the next token distribution:

$$\hat{\boldsymbol{o}}_{t+1} = \mathcal{M}_{\text{Small}}^{\text{EAGLE}}((\boldsymbol{o}_1, \cdots, \boldsymbol{o}_t), \textit{token\_embed}(y_1, \cdots, y_t)), \ \hat{y}_{t+1} \sim \text{Softmax}(\textit{LM\_head}(\hat{\boldsymbol{o}}_{t+1}))$$

The process is repeated by appending $\hat{\boldsymbol{o}}_{t+1}, \hat{y}_{t+1}$ to the inputs to auto-regressively draft tokens $\hat{y}_{t+2}$.

## 3 METHODOLOGY

### 3.1 MIXTURE OF ATTENTIONS

We begin by defining important properties of $\mathcal{M}_{\text{Small}}$ followed by detailed architectural choices.

### 3.1.1 PARTIAL OBSERVABILITY

In Markov Decision Processes (Kaelbling et al., 1998), partial observability is a common challenge where the agent does not have enough information about the true underlying state to take optimal decisions. This limitation can significantly degrade the agent's performance. Several approaches have been proposed to mitigate this, e.g., adding additional previous observations (Mnih et al., 2015). In drafting, it is important not to suffer from partial observability to draft future tokens more accurately. We extend this notion in our context with the following:

**Property 3.1** (Partial observability). *Given a ground truth function $F : \mathcal{X} \to \mathcal{Z}$, a drafter function $f : \mathcal{Y} \to \mathcal{Z}$ and an observation function $g : \mathcal{X} \to \mathcal{Y}$, such that for any $x \in \mathcal{X}$, f(g(x)) models F(x), f suffers from partial observability if g is non-injective: $\exists (x, x') \in \mathcal{X}^2, x \neq x', g(x) = g(x')$.*

We can observe that the EAGLE drafter suffers from partial observability when $F$ is $f_{\text{Large}}$, $f$ is $\mathcal{M}_{\text{Small}}^{\text{EAGLE}}$ and $\boldsymbol{o}_1, \cdots, \boldsymbol{o}_t = g(\boldsymbol{h}_{\leq t}, (y_1, \cdots, y_t))$. In other words, $\boldsymbol{o}_1, \cdots, \boldsymbol{o}_t$ is only a partial observation of the true state $(\boldsymbol{h}_{\leq t}, y_t)$ of the dynamic system (1), hindering the capacity of $\mathcal{M}_{\text{Small}}$ to predict the right tokens of $\mathcal{M}_{\text{Large}}$.

**Layer Self-Attention**    Aiming to alleviate this, our new architecture takes as input the state of the dynamic system (1). However, $\boldsymbol{h}_{\leq t}$ is a large tensor of shape $(T, L, 2E_{kv})$ where $T$ is the sequence length, $L$ the number of layers in $\mathcal{M}_{\text{Large}}$, and $E_{kv}$ the embedding size of the key and values. Therefore, we introduce Layer Self-Attention (LSA) followed by a mean aggregation operation to reduce its dimension to $(T, 2E_{kv})$ and extract the most relevant token information from every layer (Fig 1). Self-attention is performed over the layer dimension and each token is treated independently in this layer. During drafting, we have access to the past key values of all the layers, therefore, the attention mask of LSA is bidirectional/full, see Figure 2. We only perform the LSA computations *once at the start of each drafting phase*, see Appendix A.3 for a *detailed algorithm* of the information flow.

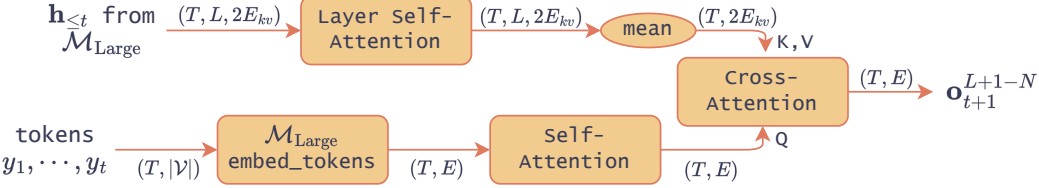

Figure 1: A schematic overview of the mixture of attentions information flow. Layer Self-Attention and mean aggregation are called *only once per drafting cycle*, i.e after each verification. New tokens are drafted auto-regressively using Self-Attention, updating only the Cross-Attention layer query.

### 3.1.2 LACK OF ON-POLICYNESS

Discrepancies between training and testing scenarios arise because, during training, transformer models are typically conditioned on ground-truth sequences, assuming that all previous inputs are correct. If this assumption seems unproblematic for the standard training of transformers, assuming it for training $\mathcal{M}_{\text{Small}}$ in speculative-decoding scenarios is much more delicate. It is known that some of the previous inputs are generated directly from $\mathcal{M}_{\text{Small}}$, therefore much less accurate. The more tokens we predict with $\mathcal{M}_{\text{Small}}$ only, the more error accumulation we can expect. To alleviate this, EAGLE adds uniform noise into its observations $(\boldsymbol{o}_1, \cdots, \boldsymbol{o}_t)$ at training time, but this is not ideal.

In order to train $\mathcal{M}_{\text{Small}}$ optimally, we need to ensure that its training and inference conditions are closely matched. Specifically, this means training $\mathcal{M}_{\text{Small}}$ as if some of the previous tokens were generated by itself. Additionally, we should account for situations where the activations from $\mathcal{M}_{\text{Large}}$ are not available, i.e. during the drafting cycle. This approach is called *on-policy* training. In on-policy training, the data used for training is generated by the same policy (or model) that is currently being trained. For example, when we train a transformer using next-token prediction on a static dataset, this is considered *off-policy* because the data doesn't change based on the model's decisions. However, if we mix this static dataset with data generated by the model itself during training, we move towards a more *on-policy* approach. Similarly, if the model won't have access to certain information, e.g. the activations of $\mathcal{M}_{\text{Large}}$ during generation, then always training $\mathcal{M}_{\text{Small}}$ with that information would also be considered *off-policy*.

However, on-policy training is very costly because we would need to generate from the model during training. To formalise this limitation, we introduce the concept of T-step boundedness:

**Property 3.2** (T-step bounded). *A drafter $f$ is said to be $T$-step bounded if, in a single forward pass, it can predict up to $T$ future tokens without additional input from $\mathcal{M}_{Large}$, i.e., $f(y_1, y_2, \ldots, y_t) \to (\hat{y}_{t+1}, \hat{y}_{t+2}, \ldots, \hat{y}_{t+T})$.*

This property is important to efficiently train the drafter. For instance, the EAGLE drafter is 1-step bounded. If one wanted to perform prediction at time $t + 2$, two forward passes would be required due to the auto-regressive layer that requires the previous $\hat{o}_{t+1}$ as input, which would be very costly to train on-policy. By contrast, a drafter that is $T$-step bounded with $T > 1$ can predict multiple future tokens within a single forward pass.

**Cross-Attention** In order to make our drafter partly $T$-step bounded with $T > 1$, the main component of our architecture is a Cross-Attention (CA) layer where the query comes from the tokens and the key and values come from $\mathcal{M}_{Large}$ activations. More precisely, the key and values come from the output of LSA. Having input queries for time $t + 1$ to $t + K$ coming into the CA layer and keys-values from $\mathcal{M}_{Large}$ only up to time $t$ effectively means the CA layer is $K$-step bounded. This allows us to train the CA layer more *on-policy* efficiently because it simulates what would happen during generation: we only have access to the activations from $\mathcal{M}_{Large}$ up to time $t$ but still have to make prediction for up to time $t + K$. Note that it is still not fully *on-policy* yet as the input queries for time $t + 1$ to $t + K$ are not assumed to be generated from $\mathcal{M}_{Small}$. During training, multiple $K$ are sampled to simulate different lengths of accepted drafts by changing the CA layer mask. For instance, in Figure 2, we have $K = 4$ followed by $K = 3$. On the contrary, during generation, we do not apply masking as we want to let $\mathcal{M}_{Small}$ attend all the currently available activations of $\mathcal{M}_{Large}$.

**Self-Attention** In order to motivate the introduction of a self-attention (SA) layer, we start by observing that the cross-attention layer is input-independent (3.3) w.r.t. the input queries, i.e one input query does not influence the results of another query.

**Property 3.3** (Input-independence). *A layer $f$ is input-independent if for any choice of $n$ inputs $\boldsymbol{x} = (x_1, \cdots, x_n)$, we have $f(\boldsymbol{x}) = (f(x_0), \cdots, f(x_n))$.*

Therefore, if the queries of the CA layer came directly from the embedded tokens $y_1, \cdots, y_t$, $\mathcal{M}_{Small}$ would not have been aware of previously drafted tokens. It would only know the previous token treated by $\mathcal{M}_{Large}$ and the most recent $y_t$. But, in order to make accurate predictions, $\mathcal{M}_{Small}$ needs to be aware of the previously drafted tokens. Hence, we introduce a causal self-attention layer on the queries to mitigate this problem, shown in Figure 1 and summarised in Table 1.

Table 1: Comparison of the properties of our new architecture.

| $\mathcal{M}_{Small}$ | Autoregressive | $T$-step bounded | More on-policy | Observability |
|---|---|---|---|---|
| Ours | SA layer | variable $T$ for CA layer | CA & LSA layers | LSA-enhanced |
| EAGLE-2 | ✓ | 1 | ✗ | partial |
| Medusa | ✗ | fixed $T$ | ✗ | partial |

### 3.2 TARGET LAYER INFERENCE

Previous work assumed that the final hidden layer before *LM_head* was the most appropriate target (activations) $\mathcal{M}_{Small}$ should predict. However, we challenge that assumption by hypothesising that targeting a deeper $\mathcal{M}_{Large}$ layer may be more advantageous in terms of draft quality. We thus decompose the dynamic system (1) layer-by-layer by introducing $l$ as the (superscript) layer index:

$$\boldsymbol{o}_{t+1}^1 = token\_embed(y_t), \qquad\qquad \boldsymbol{h}_{\leq t+1}^l, \boldsymbol{o}_{t+1}^{l+1} = f_{decoder}^l(\boldsymbol{h}_{\leq t}^l, \boldsymbol{o}_{t+1}^l),$$

$$y_{t+1} \sim Softmax(LM\_head(\boldsymbol{o}_{t+1}^{L+1})) \qquad\qquad l = 1, \ldots, L$$

where $f_{decoder}^l$ is the decoder layer of $\mathcal{M}_{Large}$ at layer $l$. The state of this new dynamic system is composed of $(\boldsymbol{o}_{t+1}^l, \boldsymbol{h}_{\leq t+1}^{\leq l}, \boldsymbol{h}_{\leq t}^{l \geq})$. We observe that to perfectly predict $\boldsymbol{o}_{t+1}^{L+1}$, it is sufficient to

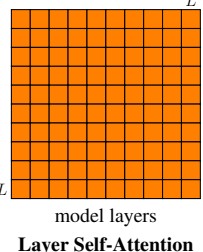 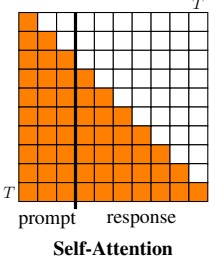 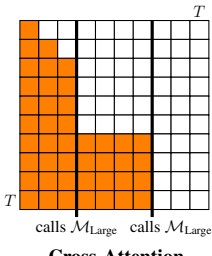

| Layer Self-Attention | Self-Attention | Cross-Attention |

Figure 2: **Layer Self-Attention**: $\mathcal{M}_{\text{Large}}$ activations are transposed so that attention is computed over the layer dimension in order to aggregate token activations across layers. **Self-Attention:** The first 3 tokens represent the prompt, speculative decoding starts at token 4. **Cross-Attention:** Tokens 4 to 7 only attend to the prompt while tokens 8 to 10 attend to the first 7 tokens once $\mathcal{M}_{\text{Large}}$ was called for the second time allowing $\mathcal{M}_{\text{Small}}$ to use the activations from the newly verified tokens.

perfectly predict $\boldsymbol{o}_{t+1}^{L}$ and *reuse* the $f_{\text{decoder}}^{L}$ of $\mathcal{M}_{\text{Large}}$ and the already computed KV cache $\boldsymbol{h}_{\leq t}^{L}$ of the layer $L$ at time $t$. The same recursive reasoning can be made to predict $\boldsymbol{o}_{t+1}^{L}$ from $\boldsymbol{o}_{t+1}^{L-1}$, etc. We assume (and later show) that predicting $\boldsymbol{o}_{t+1}^{l}$ is always easier than predicting $\boldsymbol{o}_{t+1}^{k}$ for $l < k$ due to $\boldsymbol{o}_{t+1}^{l}$ undergoing fewer layer transformations. Hence, we introduce a new hyperparameter TLI to refer to the target layer $\boldsymbol{o}^{L+1-\text{TLI}}$ that the $\mathcal{M}_{\text{Small}}$ should predict. When TLI $> 0$, the TLI last layers of $\mathcal{M}_{\text{Large}}$ (kept frozen during training) and their KV cache are used to output $\boldsymbol{o}^{L+1}$. Henceforth, we use notation (TLI $= l$) where $l$ is an integer, to denote the target layer for inference. We can now provide the equation describing our $\mathcal{M}_{\text{Small}}^{\text{Ours}}$ for a given TLI assuming $t$ was the last time we verified with $\mathcal{M}_{\text{Large}}$:

$$\hat{\boldsymbol{o}}_{T+1}^{L+1-\text{TLI}} = \mathcal{M}_{\text{Small}}^{\text{Ours}}(\boldsymbol{h}_{\leq t}, token\_embed(y_1, \cdots, y_t, \hat{y}_{t+1}, \cdots, \hat{y}_T)),$$

$$\hat{\boldsymbol{h}}_{T+1}^{l}, \hat{\boldsymbol{o}}_{T+1}^{l+1} = f_{\text{decoder}}^{l}((\boldsymbol{h}_{\leq t}^{l}, \hat{\boldsymbol{h}}_{>t,\leq T}^{l}), \hat{\boldsymbol{o}}_{T+1}^{l}), \qquad l = L - \text{TLI}, \ldots, L,$$

$$\hat{y}_{T+1} \sim \text{Softmax}(LM\_head(\hat{\boldsymbol{o}}_{T+1}^{L+1})).$$

### 3.3 Loss

Let $\mathcal{M}_{\text{Small}}$ be parameterised by $\boldsymbol{\theta}$, we use a similar training loss as EAGLE, i.e. a forward-KL loss, with a Smooth-L1 loss $\mathcal{L}$ between the predicted activations of the $\mathcal{M}_{\text{Small}}$ $\hat{\boldsymbol{o}}^{L+1-\text{TLI}}$ and the target one obtained from $\mathcal{M}_{\text{Large}}$:

$$\arg\min_{\boldsymbol{\theta}} \ \lambda_0 \text{KL}[\mathcal{M}_{\text{Large}}||\mathcal{M}_{\text{Small}}(\boldsymbol{\theta})] + \lambda_1 \mathcal{L}\left(\hat{\boldsymbol{o}}^{L+1-\text{TLI}}, \boldsymbol{o}^{L+1-\text{TLI}}\right). \tag{2}$$

To keep the training lightweight, we do not generate from $\mathcal{M}_{\text{Large}}$ or $\mathcal{M}_{\text{Small}}$ during training. This loss is only defined over the response part of the prompt of a fixed training dataset.

## 4 Experiments

In all experiments, we use *LLama3-8B-Instruct* (Dubey et al., 2024) as $\mathcal{M}_{\text{Large}}$. We train all $\mathcal{M}_{\text{Small}}$ on the Ultrachat dataset (Ding et al., 2023) without a system prompt and we do not assume that we know the system prompt at test time as it was observed that the training dataset can have a significant impact on the final performance (Yi et al., 2024). $\mathcal{M}_{\text{Small}}$ is trained with the standard Llama3-Instruct chat template. Ultrachat is composed of around 200k prompts with around 240M tokens using the LLama3 tokenizer. We use multiple test datasets for generation including various tasks such as reasoning, code generation, multi-turn conversation and summarisation. We notably relied on the SpecBench benchmark (Xia et al., 2024) and the following datasets: MT-Bench (Zheng et al., 2023), HumanEval (Chen et al., 2021), GSM8K (Cobbe et al., 2021), Alpaca (Taori et al., 2023), CNN/Daily Mail (Nallapati et al., 2016) and Natural Questions (Kwiatkowski et al., 2019). We describe additional hyperparameters and experimental settings in Appendix A.1.

We compare our method to EAGLE-2 and an independent distilled $\mathcal{M}_{\text{Small}}$ of similar size (denoted "Independent"). In order to train the EAGLE model, we assume TLI $= 0$ in the distillation loss (2).

The independent $\mathcal{M}_{\text{Small}}$ leverages the *token_embed* and *LM_head* parameters of $\mathcal{M}_{\text{Large}}$ with only the decoder layers trained using an identical distillation loss (2) and $\lambda_1 = 0$. We do not compare to Medusa as EAGLE has consistently demonstrated superior speedups on various benchmarks (Xia et al., 2024). We also compare the performance of the official EAGLE-2 weights shared by Li et al. (2024b). We refer to this as "EAGLE-2 off.". Note that this model was trained on different data and with a fixed system prompt. We take care to match the number of model parameters, i.e. "Ours (N=0)", "EAGLE-2", "EAGLE-2 off.", "Independent 1.3B" and "Glide" all have 1.3B parameters (250M trainable and 1.05B frozen, for the "LM head" and "token embed" layers). We chose 250M trainable parameters to be directly comparable to EAGLE-2 and their official checkpoint. For tree decoding, we use a max breadth of 8, a depth of 6 and 62 max tokens to verify. We use float16 except for the attention softmax weights that are upscaled to float32.

We use standard metrics: *token-per-second* and *speedup ratios* to measure walltime improvements as well as hardware-independent metrics: average *acceptance length* $\tau$ (the average number of $\mathcal{M}_{\text{Small}}$ tokens accepted by $\mathcal{M}_{\text{Large}}$) and the *number of calls* to $\mathcal{M}_{\text{Large}}$.

## 4.1 SINGLE DEVICE

We now present the main single-device experiments using the SpecBench Xia et al. (2024) benchmark without a system-prompt to ensure a fair comparison between models.

Table 2: Speedup ratio and acceptance length $\tau$ on SpecBench using prompts from MT-Bench, HumanEval, GSM8K, Alpaca, Sum and QA datasets. Each model is fine-tuned for 30 epochs and uses EAGLE-2 tree decoding.

| $\mathcal{M}_{\text{Small}}$ | Total size | Trainable size | MT-bench Speedup | $\tau$ | HumanEval Speedup | $\tau$ | GSM8K Speedup | $\tau$ | Alpaca Speedup | $\tau$ | CNN/DM Speedup | $\tau$ | Natural Ques. Speedup | $\tau$ | Mean Speedup | $\tau$ |
|---|---|---|---|---|---|---|---|---|---|---|---|---|---|---|---|---|
| Ours (TLI=3) | 1.8B | 250M | 1.74 | **4.65** | 2.02 | **5.41** | 1.74 | **4.65** | 1.81 | **4.80** | 1.89 | **5.04** | 1.59 | **4.23** | 1.79 | **4.79** |
| Ours (TLI=1) | 1.55B | 250M | **1.83** | 4.19 | **2.29** | 5.30 | **1.83** | 4.19 | 2.02 | 4.65 | 2.04 | 4.74 | 1.71 | 3.94 | **1.95** | 4.50 |
| Ours (TLI=0) | 1.3B | 250M | 1.80 | 3.86 | 2.28 | 4.98 | 1.80 | 3.86 | **2.03** | 4.36 | **2.10** | 4.55 | **1.72** | 3.73 | **1.95** | 4.22 |
| EAGLE-2 | 1.3B | 250M | 1.77 | 3.55 | 1.95 | 3.92 | 1.69 | 3.36 | 1.89 | 3.77 | 1.84 | 3.69 | 1.66 | 3.32 | 1.78 | 3.60 |
| EAGLE-2 off. | 1.3B | 250M | 1.75 | 3.52 | 2.06 | 4.15 | 1.80 | 3.60 | 1.70 | 3.37 | 1.60 | 3.19 | 1.38 | 2.75 | 1.71 | 3.43 |
| Independent | 1.7B | 650M | 1.50 | 3.63 | 1.91 | 4.64 | 1.26 | 3.01 | 1.57 | 3.81 | 1.56 | 3.78 | 1.72 | 3.94 | 1.58 | 3.80 |
| Independent | 1.3B | 250M | 1.23 | 3.50 | 1.50 | 4.36 | 0.95 | 2.70 | 1.33 | 3.79 | 1.28 | 3.59 | 1.10 | 3.13 | 1.23 | 3.51 |
| Glide | 1.3B | 250M | 1.69 | 3.62 | 2.06 | 4.43 | 1.54 | 3.27 | 2 | 4.27 | 1.6 | 3.37 | 1.59 | 3.41 | 1.74 | 3.72 |

Looking at Table 2, we can see that our Mixture of Attentions for SD achieves SOTA speedups when TLI = 1 and TLI = 0. Compared to EAGLE-2, we are on average 9.5% faster in terms of *tokens-per-second* generated. We also increase the acceptance length by 25% when $N = 1$. More single device experiments e.g. on the full HumanEval dataset are shown in Appendix A.4.

## 4.2 CLIENT-SERVER

In this study, we investigate how *self-drafting* with our method performs in a client-server scenario. To do so, we place $\mathcal{M}_{\text{Small}}$ on a client device and host $\mathcal{M}_{\text{Large}}$ on a server (see Appendix A.2 for an illustration). The server is performing verification and sends the relevant $\mathcal{M}_{\text{Large}}$ activations to the client, which in turn is proposing new tokens. The server has 3 times more float16 tflops than the client. The devices are located in two different cities, separated by ∼300 km. The ping between the devices is around 9 ms and the bandwidth ∼50 Mbits/sec. In order to simulate a realistic client-server scenario, we are using 5G and 4G network profiles. In 4G, we assume a maximum of 20 Mbits/sec with a normally distributed delay of 21 ms ± 19 ms and a 0.1% chance of dropping packets. In 5G, we assume a normally distributed delay of 10 ms ± 10 ms with a 0.1% chance of dropping packets. To do so, we rely on the Linux traffic control subsystem.

In this scenario, the token-per-second performance also depends on the size of the messages. To this end, we analyse the length of the messages sent between the client and the server (see Table 7). There is a clear distinction between *self-drafting* methods that need to send/receive activation tensors and *independent* methods that only exchange text (e.g. token ids). Therefore, we shall analyse whether the improvement in drafting quality can offset the increase in message lengths. On the client, we encode each node in the draft tree using 3 bytes for the token id and 1 byte for its position in the tree. The server answers with the accepted tokens encoded using 3 bytes each plus the associated

activations, if required. For Llama3-8B-Instruct and $N \leq 1$, our architecture's payload is less than or equal to EAGLE message lengths. In order to further reduce message sizes, we quantise the $E$ and $E_{kv}$ tensors to 8 bits. For both EAGLE and Mixture of Attentions, the initial message sent by the server (before the first token is drafted) is typically the biggest as it represents the activations of the entire prompt. Therefore, we additionally gzip-compress this message after quantisation.

Table 3: Performance on *HumanEval* with EAGLE-2 tree decoding under 5G and 4G profiles.

| $\mathcal{M}_{\text{Small}}$ | Total size | Trainable size | Tokens per second ↑ | | Acceptance length ↑ | Calls $\mathcal{M}_{\text{Large}}$ ↓ |
|---|---|---|---|---|---|---|
| | | | 5G | 4G | | |
| Ours (TLI=3) | 1.8B | 250M | 25.0 | 14.6 | **4.99** | **20.8** |
| Ours (TLI=1) | 1.55B | 250M | 30.6 | 20.3 | 4.68 | 22.5 |
| Ours (TLI=0) | 1.3B | 250M | **34.1** | **25.1** | 4.30 | 24.1 |
| EAGLE-2 | 1.3B | 250M | 24.3 | 13.6 | 2.81 | 36.4 |
| EAGLE-2 off. | 1.3B | 250M | 28.6 | 15.0 | 3.51 | 29.5 |
| Independent | 1.7B | 650M | 28.5 | 23.7 | 3.73 | 27.1 |
| Independent | 1.3B | 250M | 18.3 | 16.1 | 3.16 | 32.4 |

In Table 3, we can observe that "Ours (TLI=0)" achieves the fastest decoding speeds. Interestingly, it is even faster than independent small models that do not exchange any activation tensors. As expected, our Mixture of Attentions is not as fast as in the single device setting, but it can recover the speed of vanilla decoding in a single device setup (33 tokens-per-second, see Appendix A.4).

However, for this setting to be viable, just recovering the speed of vanilla decoding is not sufficient as it does not provide an advantage over an API call to $\mathcal{M}_{\text{Large}}$. Therefore, we show that our model can continue to generate the response by simulating a complete disconnection from the server.

Table 4: The success rate (pass@1, greedy decoding) on *HumanEval* in the event of an interrupted connection between the client and the server. EAGLE-2 tree decoding is used.

| | | | A disconnection occurs after B new tokens. | | | | |
|---|---|---|---|---|---|---|---|
| $\mathcal{M}_{\text{Small}}$ | Total size | Trainable size | B = 1 | B = 10 | B = 25 | B = 50 | B = ∞ |
| Ours (TLI=3) | 1.8B | 250M | 2.48 % | **11.18 %** | 18.01 % | **31.67 %** | 45.9 % |
| Ours (TLI=1) | 1.55B | 250M | **3.10 %** | 10.55 % | **21.11 %** | 30.43 % | 45.9 % |
| Ours (TLI=0) | 1.3B | 250M | 2.48 % | 9.31 % | 19.2 % | 29.81 % | 45.9 % |
| EAGLE-2 | 1.3B | 250M | 0 % | 8.07 % | 16.77 % | 27.32 % | 45.9 % |
| EAGLE-2 off. | 1.3B | 250M | 1.24 % | 6.83 % | 18.01 % | 28.57 % | 45.9 % |
| Independent | 1.7B | 650M | 0 % | 6.83 % | 18.63 % | 29.81 % | 45.9 % |
| Independent | 1.3B | 250M | 0 % | 6.21 % | 18.01 % | 27.95 % | 45.9 % |
| | | | Generation stops after B new tokens. | | | | |
| Without local model (lower bound) | | | 0 % | 5.59 % | 16.77 % | 27.32 % | 45.9 % |

In Table 4, we can see that indeed, if a disconnection occurs, unlike API calls to $\mathcal{M}_{\text{Large}}$, we can continue to generate the response *right on the device*, i.e. complete additional correct solutions to competitive programming problems in HumanEval. Therefore, with an acceptable speed and the possibility to generate useful responses after a disconnection, we prove the viability of our proposed client-server setting, paving the way for a new framework for serving LLMs with small devices.

## 4.3 ABLATION STUDY

We now present important ablation results for different components of our Mixture of Attentions architecture. Since multiple models were required to be fine-tuned for this study, we have limited each run to 10 epochs. For this ablation, we introduce the "Ours (TLI=$l$, -LSA)" variant that does not rely on LSA and takes as input $o_1, \cdots, o_t$ as the keys and values of the CA layer. We also include two more EAGLE baselines, one with additional trainable parameters "EAGLE (more params)" and another with additional decoder layers "EAGLE (more layers)" but an equal number of trainable

parameters. This is to ensure that the benefit of our architecture does not come from simply adding decoder layers or parameters. In this experiment, we use the HumanEval dataset with strict stopping criteria, exiting decoding as soon as the model no longer generates source code.

Table 5: An ablation study of our proposed architecture, tested on *HumanEval*. Each model is trained on $\sim$2.4B tokens. Chain (not tree) drafting with maximum 4 tokens is used for this study. The averages are computed over around 8500 drafting-verification cycles.

| $\mathcal{M}_{\mathrm{Small}}$ | Total size | Trainable size | Tokens per second | Acceptance length ($\tau$) |
|---|---|---|---|---|
| Ours (TLI=3) | 1.8B | 250M | 39 | **2.54** |
| Ours (TLI=1) | 1.55B | 250M | 39 | 2.25 |
| Ours (TLI=0) | 1.3B | 250M | **40** | 2.14 |
| Ours (TLI=1, -LSA) | 1.55B | 250M | 21 | 1.28 |
| Ours (TLI=1, -LSA, $o_1, \cdots o_t$ inputs) | 1.55B | 250M | 36 | 2.04 |
| Ours (TLI=0, -LSA, $o_1, \cdots o_t$ inputs) | 1.3B | 250M | 38 | 1.93 |
| EAGLE | 1.3B | 250M | 30 | 1.45 |
| EAGLE (more params) | 1.45B | 400M | 29 | 1.28 |
| EAGLE (more layers) | 1.3B | 250M | 27 | 1.01 |

**Does the on-policyness (brought with the CA layer) and the $T$-step bounded property have a positive impact on the quality of the drafts?**  In Table 5, we compare EAGLE with "Ours (TLI=0, -LSA)" for an answer to this question. We can see that these components provide a major improvement of 26% in tokens-per-second as well as improved acceptance length of 33%.

**How does partial observability influence the drafter *acceptance rate*?**  In Table 5, we can compare "Ours (TLI=0, -LSA)" to "Ours (TLI=0)" as well as "Ours (TLI=1, -LSA)" to "Ours (TLI=1)" and report that the tokens-per-second performance improves by 6% by introducing LSA, decreasing partial observability. Its impact is less crucial than the on-policyness brought by the CA layer.

**Does increasing TLI increase the *acceptance rate*?**  Finally, by looking at the variation of TLI in Tables 2,3 and 5, increasing TLI also increases the acceptance length, as we hypothesised. However, this does not always have a positive impact on the tokens-per-second rate as it also increases the computational time of drafting. In the event of a complete disconnection in a client-server setting, however, a higher TLI will improve the quality of responses, which is something to consider when deploying Mixture of Attentions for SD on mobile devices.

## 5  RELATED WORK

Medusa (Cai et al., 2024a) is one of the earliest works leveraging the activations of $\mathcal{M}_{\mathrm{Large}}$ as inputs to $\mathcal{M}_{\mathrm{Small}}$ for the purpose of SD. Thanks to their work, speculative decoding can be applied to any LLM by distilling an $\mathcal{M}_{\mathrm{Small}}$. It generates $K$ future tokens in parallel by training $K$ new *LM_heads* where each head predicts a token at position $k \in K$ (Gloeckle et al., 2024). It was later extended by Kim et al. (2025) by refining the block drafts using task-independent n-gram and neural language models as lightweight rescorers. EAGLE (Li et al., 2024c) and Hydra (Ankner et al., 2024) are auto-regressive extensions of Medusa. They observe that non-auto-regressive generation limits the acceptance length as $\mathcal{M}_{\mathrm{Small}}$ is not aware of previous tokens. We do not compare to Medusa or Hydra as EAGLE is ranked higher on the SpecBench leaderboard.

Tandem Transformers (Nair et al., 2024) propose an effective integration of $\mathcal{M}_{\mathrm{Large}}$ and $\mathcal{M}_{\mathrm{Small}}$ by letting $\mathcal{M}_{\mathrm{Small}}$ attend to the down-projected hidden states of $\mathcal{M}_{\mathrm{Large}}$. These rich contextualised representations enable $\mathcal{M}_{\mathrm{Small}}$ to draft hypotheses with a higher acceptance rate as the two models are aligned on shared hidden states. We were not able to compare with them because of the lack of open-source implementation, the use of closed-source LLMs and an undisclosed amount of data/compute to reproduce the work. Moreover, tandem transformers appear to have a high communication over-head between big and small models, making it unrealistic for a client/server setting.

Orthogonal to our work, researchers have recently proposed *training-free* SD methods. Lookahead Decoding (Fu et al., 2024) generates new tokens with a single $\mathcal{M}_{\mathrm{Large}}$ using Jacobi iterations, extended by CLLM Kou et al. (2024) and Ouroborous (Zhao et al., 2024). We evaluated the latter in

our settings, however, it was shown to be less efficient than the EAGLE-2 tree decoding strategy, see Appendix A.4. For additional related and orthogonal work in the extended SD landscape, we refer the reader to Xia et al. (2024) for a detailed and highly informative speculative decoding survey.

Du et al. (2024) previously proposed to leverage the KV-cache of some layers of $\mathcal{M}_{\text{Large}}$. They do not theoretically justify why using the KV-cache instead of the output of each layer, nor how to exactly choose which layer to include as input of $\mathcal{M}_{\text{Small}}$. However, with our dynamical system point of view, we showed that the KV-cache of all the layers is part of the state. The introduction of LSA allows to exploit it in its whole with a limited number of layers, whereas Du et al. (2024) would need to have the same number of layers in $\mathcal{M}_{\text{Small}}$ and $\mathcal{M}_{\text{Large}}$ to fully capture it, resulting in a slow drafting speed.

Although we focused on improving the current SOTA method (EAGLE-2), our observations (partial observability, on-policyness and target inference layer) are true for many self-drafting methods, for instance, it could also be applied to Medusa (Cai et al., 2024a), MLP Speculator (Wertheimer et al., 2024) or Gloeckle et al. (2024); Kim et al. (2025).

Regarding non-self-drafting SD, it should be studied on a case-by-case basis. For instance, target inference layer could potentially be applied to independent small models. Many student-teacher distillation frameworks (Gu et al., 2024; Zhou et al., 2023), already leverage the on-policyness property by generating directly from the student but are mostly are 1-step bounded (therefore expensive to train). For SD methods based on lookahead decoding, it would generally not apply. One exception is Ouroboros (Zhao et al., 2024) that leverages a small model with lookahead decoding. Their small model could also benefit from our solutions.

## 6 CONCLUSION

We have introduced a Mixture of Attentions architecture for Speculative Decoding to effectively address several limitations of existing state-of-the-art methods. In order to enhance drafting accuracy of $\mathcal{M}_{\text{Small}}$, we proposed a mixture of attention layers: Layer Self-Attention to mitigate partial observability and Self-Attention followed by Cross-Attention to train more on-policy. We have then introduced Target Layer Inference, a novel method that lets $\mathcal{M}_{\text{Small}}$ reuse the last $N$ layers of $\mathcal{M}_{\text{Large}}$, enabling a trade off between the drafting speed and accuracy. Experimental results show that we achieve state-of-the-art decoding speedups in the standard single-device setup, improving over EAGLE-2 by 9.5% and extending acceptance lengths by up to 25%. We have also introduced a client-server paradigm and demonstrated that our *self-drafting* speculative decoding method is a viable alternative to API calls to $\mathcal{M}_{\text{Large}}$. Under this paradigm, the client can continue to generate responses with the highest accuracy and speed after a complete disconnection from the network. As a future direction, it would be interesting to investigate whether $N$ could be predicted by $\mathcal{M}_{\text{Small}}$ to automatically balance speed and accuracy depending on the current network conditions.

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

## A APPENDIX

The source code is publicly available at `https://github.com/huawei-noah/HEBO/tree/mixture-of-attentions/`.

### A.1 HYPERPARAMETERS

Table 6: List of our hyperparameters.

| **Distillation** | |
| --- | --- |
| Learning rate for gradient descent | $3 \cdot 10^{-5}$ |
| Total numbers of transformer updates | 186000 |
| Minibatch size | 32 |
| Mixed-precision training | yes, float16 |
| Weight of reserve KL loss ($\lambda_0$) | 0.1 |
| Weight of L1 smooth loss ($\lambda_1$) | 1.0 |
| L2 gradient clipping | 1.0 |
| $T$-step bounded mask for the CA layer | Uniform between 5 to 15 |
| **Architecture** | |
| Number of layers $L$ of $\mathcal{M}_{\text{Large}}$ | 32 |
| Embedding dimension $E$ of $\mathcal{M}_{\text{Large}}$ | 4096 |
| Embedding dimension of keys and values $E_{kv}$ of $\mathcal{M}_{\text{Large}}$ | 1024 |
| Dropout rate | 0.0 |
| Embedding dimension of Layer Self-Attention | 2048 |
| Embedding dimension of Self-Attention | 4096 |
| Embedding dimension of Cross-Attention | 4096 |
| Size of the MLP projection after Layer Self-Attention | 6144 |
| Size of the MLP projection after Self-Attention | 512 |
| Size of the MLP projection after Cross-Attention | 7168 |
| Embedding dimension of keys and values of Layer Self-Attention | 1024 |
| Embedding dimension of keys and values of Self-Attention | 512 |
| Embedding dimension of keys and values of Cross-Attention | 1024 |

### A.2 CLIENT SERVER DEPLOYMENT

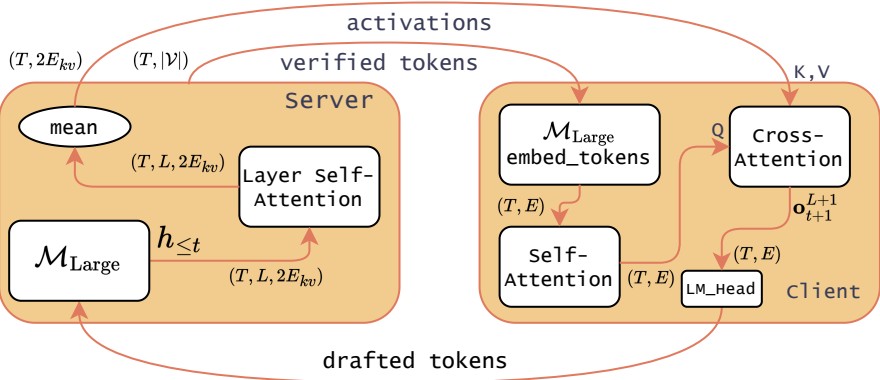

Figure 3: A client-server setting for our mixture of attentions architecture with $N = 0$.

Table 7: The size of the message (before quantisation) in bytes. $M$ = number of nodes in the draft tree, $A$ = number of accepted tokens, $E$ = hidden size, $E_{kv}$ = hidden size of key and query vectors.

| $\mathcal{M}_{\text{Small}}$ | Sent by Client | Sent by Server |
|---|---|---|
| Ours | $4M$ | $3A + 2AE_{kv}(\text{TLI} + 1)$ |
| EAGLE | $4M$ | $3A + AE$ |
| Independent | $4M$ | $3A$ |

## A.3 ALGORITHM

---

**Algorithm 1** Generation algorithm for $\mathcal{M}_{\text{Small}}^{\text{Ours}}$ assuming chain decoding

---

**Require:** Input sequence $\boldsymbol{y} = (y_1, y_2, \ldots, y_t)$, draft length $K$, target layer inference TLI
1: Obtain $\boldsymbol{h}_{\leq t}$ activations and $y_{t+1}$ with a forward pass in $\mathcal{M}_{\text{Large}}$ given input $\boldsymbol{y}$
2: $\boldsymbol{y} \leftarrow (\boldsymbol{y}, y_{t+1})$
3: $\boldsymbol{kv} \leftarrow \text{LSA\_layer\_with\_mean}\,(\boldsymbol{h}_{\leq t})$
4: **while** stopping criteria is not meet on $\boldsymbol{y}$ **do**
5:    **for** $i = 1$ to $K$ **do**
6:       $\boldsymbol{q} \leftarrow \text{SA\_layer}(\text{token\_embed}(\boldsymbol{y}))$
7:       $\hat{\boldsymbol{o}}^{L+1-N} \leftarrow \text{CA\_layer}(\boldsymbol{q}, \boldsymbol{kv})$
8:       **if** $N > 0$ **then**
9:          **for** $l = L - N$ to $L$ **do**
10:             $[\hat{\boldsymbol{h}}^l, \hat{\boldsymbol{o}}^{l+1}] \leftarrow f_{\text{decoder}}^l((\boldsymbol{h}_{\leq t}^l, \hat{\boldsymbol{h}}_{>t, \leq t+i}^l), \hat{\boldsymbol{o}}^l)$
11:          **end for**
12:       **end if**
13:       $\hat{y} \sim \text{Softmax}(\textit{LM\_head}(\hat{\boldsymbol{o}}^{L+1}))$
14:       $\boldsymbol{y} \leftarrow (\boldsymbol{y}, \hat{y})$
15:    **end for**
16:    Identify $K'$ verified tokens out of the $K$ latest tokens of $\boldsymbol{y}$, obtain associated $\boldsymbol{h}'$ and obtain $y'$ with a forward pass in $\mathcal{M}_{\text{Large}}$ with inputs $\boldsymbol{y}_{|\boldsymbol{y}|-K, \cdots, |\boldsymbol{y}|}$ and $\boldsymbol{h}_{\leq t}$
17:    $\boldsymbol{kv}' \leftarrow \text{LSA\_layer\_with\_mean}\,(\boldsymbol{h}')$
18:    $\boldsymbol{kv} \leftarrow (\boldsymbol{kv}, \boldsymbol{kv}')$
19:    Update $\boldsymbol{h}$ by appending the new $\boldsymbol{h}'$ components
20:    Discard previous $\hat{\boldsymbol{h}}$
21:    $\boldsymbol{y} \leftarrow \boldsymbol{y}_{1, \cdots, |\boldsymbol{y}|-K+K'}$ (keep only the verified tokens)
22:    $t \leftarrow |\boldsymbol{y}|$
23:    $\boldsymbol{y} \leftarrow (\boldsymbol{y}, y')$
24: **end while**
25: **return** $\boldsymbol{y}$

---

## A.4 ADDITIONAL EXPERIMENTS

**Accuracy of the generated text** We ran several experiments to assess the quality of the generated responses using greedy decoding. We focused on 3 datasets from SpecBench (HumanEval, GSM8K and CNN/DM) that do not require access to proprietary models/APIs for evaluation (llm-as-a-judge).

Table 8: Quality of the generated text.

| Vanilla decoding | HumanEval (pass@1) | GSM8K (accuracy) | CNN/DM (Rouge-L f-score) |
|---|---|---|---|
| Llama3-8B-Instruct | 62.5% | 80% | 0.3071 |
| Speculative Decoding | HumanEval (pass@1) | GSM8K (accuracy) | CNN/DM (Rouge-L f-score) |
| Ours (TLI=3) | 62.5% | 80% | 0.3053 |
| Ours (TLI=1) | 62.5% | 81.25% | 0.3068 |
| Ours (TLI=0) | 62.5% | 80% | 0.3070 |
| EAGLE-2 | 62.5% | 81.25% | 0.3062 |
| EAGLE-2 off | 62.5% | 80% | 0.3056 |
| Independent 1.7B | 62.5% | 80% | 0.3067 |
| Independent 1.3B | 62.5% | 80% | 0.3064 |

We report the results in Table 8. The pass@1 on HumanEval is the same across all methods. The accuracy on GSM8K actually improves w.r.t the base model on one question for Ours (TLI=1) and EAGLE-2. Finally, the ROUGE scores are also extremely similar, leading us to conclude that any differences to the base model are negligible and almost certainly appear due to using float16.

**Qwen2.5 3B**  We trained 3 additional small models on the Ultrachat dataset to accelerate Qwen2.5 3B. EAGLE recommends to use one decoder layer of the big LLM to define the size of the small LM, which leads to a trainable size of 80M parameters. We kept the shared "embed_tokens/LM_head" layer frozen.

Table 9: Speedup ratio and acceptance length $\tau$ on SpecBench using prompts from MT-Bench, HumanEval, GSM8K, Alpaca, Sum and QA datasets with Qwen2.5-3B Instruct.

| $\mathcal{M}_{Small}$ | Total size | Trainable size | MT-bench Speedup | $\tau$ | HumanEval Speedup | $\tau$ | GSM8K Speedup | $\tau$ | Alpaca Speedup | $\tau$ | CNN/DM Speedup | $\tau$ | Natural Ques. Speedup | $\tau$ | Mean Speedup | $\tau$ |
|---|---|---|---|---|---|---|---|---|---|---|---|---|---|---|---|---|
| Ours (TLI=0) | 0.4B | 80M | **1.71** | **3.72** | **2.18** | **4.76** | **1.60** | **3.46** | **1.88** | **4** | **1.78** | **3.89** | **1.68** | **3.59** | **1.80** | **3.9** |
| EAGLE-2 | 0.4B | 80M | 1.59 | 3.2 | 1.84 | 3.70 | 1.53 | 3.06 | 1.81 | 3.54 | 1.60 | 3.23 | 1.62 | 3.17 | 1.66 | 3.31 |
| Independent | 0.4B | 80M | 1.59 | 3.37 | 2.04 | 4.38 | 1.44 | 3.03 | 1.70 | 3.52 | 1.54 | 3.27 | 1.50 | 3.12 | 1.63 | 3.44 |

**Higher batch size with vLLM**  We implemented our approach in vLLM (Kwon et al., 2023) without tree decoding to support higher batch sizes and continuous batching.

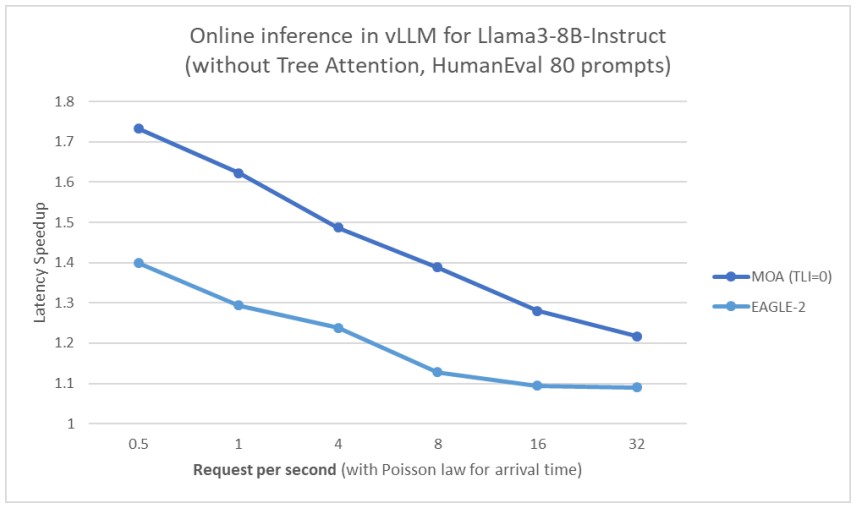

Figure 4: vLLM inference with continuous batching.

**HumanEval in single device**  To perform this experiment, we reuse the same full HumanEval dataset with a strict stopping criteria as done in the ablation study in the single device setting.

Table 10: Test on *Human Eval*, each model is trained for 30 epochs.

| $\mathcal{M}_{\text{Small}}$ | Decoding | Total size | Trainable size | Tokens per second | Acceptance length ($\tau$) |
|---|---|---|---|---|---|
| Ours (TLI=3) | EAGLE-2 | 1.8B | 250M | 54 | **5.02** |
| Ours (TLI=1) | EAGLE-2 | 1.55B | 250M | **58** | 4.70 |
| Ours (TLI=0) | EAGLE-2 | 1.3B | 250M | 57 | 4.30 |
| EAGLE | EAGLE-2 | 1.3B | 250M | 43 | 2.82 |
| EAGLE off. | EAGLE-2 | 1.3B | 250M | 52 | 3.50 |
| Independent | EAGLE-2 | 1.7B | 650M | 46 | 3.72 |
| Independent | EAGLE-2 | 1.3B | 250M | 34 | 3.17 |
| Independent | Ouroboros | 1.7B | 650M | 39 | 2.37 |
| Baseline | Vanilla | - | - | 33 | 1 |

From Table 10, we can observe we are 26% faster than EAGLE/EAGLE-2. We are also faster than independent small models and Ouroboros (Zhao et al., 2024).

## A.5 COMPLEXITY ANALYSIS

Let us analyze the standard decoder-only transformers doing vanilla decoding:

- in the first prefill stage, it grows in $\mathcal{O}(LKE(E+K))$ given we have $L$ self-attention layers with $K$ input tokens and an embedding size of $E$

- for the K' new decoded tokens, it grows in $\mathcal{O}(\sum_i^{K'} L(E^2 + E(K+i))) = \mathcal{O}(LE(EK' + KK' + K'^2))$.

If we assume $E$ and $L$ are fixed, it grows in $\mathcal{O}((K + K')^2)$ overall. For speculative decoding, the first prefill stage is the same. Assuming $S$ tokens are verified at a time, the verification would grow in $\mathcal{O}(\sum_i^{\frac{K'}{S}} L(SE^2 + SE(K+i))) = \mathcal{O}(LE(EK' + KK' + K'^2))$, leading to the same complexity as vanilla decoding. It dominates the complexity of self-drafting, but we can still analyse it. For EAGLE, decoding a new token grows in $\mathcal{O}(E^2 + EK)$ as it is a single self-attention layer. For our Mixture of Attentions architecture, the Self-Attention and Cross-Attention layers also grow in $\mathcal{O}(E^2 + EK)$. The Layer-Self Attention is only called once after every verification stage, so not at every decoding step, it grows in $\mathcal{O}(ALE_{kv}^2 + AE_{kv}L^2)$ if $A$ is the number of accepted tokens in the previous phase. In our experiments, if we look at the first term, $ALE_{kv}^2$ is smaller than $number\_of\_decoded\_tokens \times E^2$ as $E_{kv}$ is 4 times smaller than $E$, $L$ is 32, $A$ is in average 4.5 and $number\_of\_decoded\_tokens$ is 48. Similarly for the second term, $AE_{kv}L^2$ is usually smaller than $number\_of\_decoded\_tokens \times EK$ as soon as the request contains more than 24 tokens. Therefore, the time complexity is the same as EAGLE overall.

## A.6 PRIVACY APPLICATION

Another advantage of the client-server setup is that we can selectively ensure privacy for the client by only sending the non-sensitive part of the prompt to the server. Essentially, the client can split their input into a consecutive "safe" text and a "private" text. The server processes only the "safe" text, which could be general context or non-sensitive information. The client keeps the "private" text, such as confidential data or sensitive instructions, and handles this part locally with $\mathcal{M}_{\text{Small}}$.

For instance, the client might send the server some Python code along with a general description. However, any sensitive information, such as the login and password to inject into the code, remains on the client side and is not transmitted to the server. It is only passed to $\mathcal{M}_{\text{Small}}$. This approach leverages the activations of $\mathcal{M}_{\text{Large}}$ to increase the accuracy of $\mathcal{M}_{\text{Small}}$ for parts of the task while ensuring that sensitive information is never exposed outside the client's environment.

