# OpenReview forum: "Mixture of Attentions For Speculative Decoding"
_ICLR.cc/2025/Conference — ICLR 2025 Poster_

### Official Review · Reviewer_Sr5b · 2024-10-26

**Soundness:** 3
**Presentation:** 2
**Contribution:** 3
**Rating:** 6
**Confidence:** 4

**Summary:**

The paper proposes an improved architecture for speculative decoding, which incorporates Layer Self-Attention and Cross-Attention mechanisms to address common limitations like partial observability and lack of on-policy training. Experimental results demonstrate that this approach achieves state-of-the-art speed and accuracy in single-device and client-server deployments, maintaining high accuracy even during disconnections. The limitation analysis of SD in the paper provides insights into this field.

**Strengths:**

(1) The limitation analysis of SD in the paper is very interesting, it links the limitation of SD (e.g., the SD may still give different predictions compared to only using the large model even when the previous tokens are all accepted ) with theoretical analysis, i.e., partial observability. This may provide further insights of optimizing SD algorithms.

(2) It is a very good point that the paper design and evaluate the scenario that the small model is on a resource-limited client and the large model is on the server. It is a very realistic scenario to use SD.

(3) The paper uses SOTA models such as Llama-3 for evaluation. Also the benchmarks used (e.g., HumanEval, MT-Bench) are challenging enough for evaluation.

**Weaknesses:**

(1) The link between the limitation of SD and the proposed attention-based method is not clear enough. Why Layer Self-Attention can solve partial observability? Further intuitive explanations or theoretical analyses are needed.

(2) Authors mention that their work is based on a SOTA SD method named EAGLE. Then my concern is, is the proposed algorithm transferable to other SD algorithms? For example, can the algorithm be applied to non-self-draft SD? The point here is that, if the algorithm can be applied to most of the SD frameworks, there will be more contribution. Otherwise, it's just an optimization of one of the previous works.

(3) No accuracy is displayed in the experiment section. I know that this is common in SD papers. However, when I was trying SD codes, I found that the SD performance is usually not as good as the large model, which is the purpose of SD. Thus, because this paper aims to solve problems such as partial observability, will it also increase the performance of SD, in addition to the efficiency? I believe experiments related to this point will make the paper more convincing.

**Questions:**

Please refer to the weakness.

---

> ### Author Response · Authors · 2024-11-20
>
> Thank you very much for your feedback and suggestions. We are delighted that you found the discussion, analysis and evaluation, particularly the client-server use case, valuable and interesting. Let us address your suggestions that will help us improve the paper further.
>
> > **Q1: The link between the limitation of SD and the proposed attention-based method is not clear enough. Why Layer Self-Attention can solve partial observability? Further intuitive explanations or theoretical analyses are needed.**
>
> The primary reason is that, with Layer Self-Attention, we can observe the whole dynamical system state whereas EAGLE or Medusa would only observe a limited (transformed) subset of this state.
>
> > **Q2: Authors mention that their work is based on a SOTA SD method named EAGLE. Then my concern is, is the proposed algorithm transferable to other SD algorithms? For example, can the algorithm be applied to non-self-draft SD? The point here is that, if the algorithm can be applied to most of the SD frameworks, there will be more contribution. Otherwise, it's just an optimization of one of the previous works.**
>
> Thank you for raising this very interesting question.
> Although we focused on improving the current SOTA method (EAGLE-2), **our observations (partial observability, on-policyness and target inference layer) are true for many self-drafting** methods, for instance, it could also be applied to MEDUSA, MLP Speculator (https://arxiv.org/abs/2404.19124) or [4, 5].
> We will make this clearer in the final version of the paper.
>
> Regarding non-self-drafting SD, this should be studied on a case-by-case basis.
> For instance, target inference layer could potentially be applied to independent small models.
> Many student-teacher distillation frameworks [1-2], already leverage the on-policyness property by generating directly from the student but are mostly are 1-step bounded (therefore expensive to train).
> For SD methods based on lookahead decoding, it would generally not apply.
> One exception is Ouroboros [3] that leverages a small model with lookahead decoding.
> Their small model could also benefit from our solutions.
>
> [1] Gu, Yuxian, et al. Knowledge distillation of large language models. ICLR, 2024.
>
> [2] Zhou, YYongchao, et al. Distillspec: Improving speculative decoding via knowledge distillation. ICLR 2024.
>
> [3] Zhao, Weilin, et al. "Ouroboros: Speculative Decoding with Large Model Enhanced Drafting." EMNLP 2024.
>
> [4] Gloeckle et al. (2024), Better \& Faster Large Language Models via Multi-token Prediction.
>
> [5] Kim et al. (2024), Accelerating Blockwise Parallel Language Models with Draft Refinement.
>
> > **Q3: No accuracy is displayed in the experiment section. I know that this is common in SD papers. However, when I was trying SD codes, I found that the SD performance is usually not as good as the large model, which is the purpose of SD. Thus, because this paper aims to solve problems such as partial observability, will it also increase the performance of SD, in addition to the efficiency? I believe experiments related to this point will make the paper more convincing.**
>
> In Table 3, we show (accuracy-like) pass@k scores on HumanEval in a client-server setting, as an example of performance after a partial/full disconnection. Tables 2 and 3 are focused on efficiency, citing standard SD metrics such as acceptance length and speedup ratios. This practice is very common for SD papers as the verification step of SD is theoretically shown to preserve the output distribution of the large LLM.
>
> The difference you observed between SD and non-SD methods is probably due to how verification is implemented. Therefore, mitigating the partial observability will not overcome this problem as it does not touch the verification algorithm.
> If you observed a difference in greedy decoding, it is likely due to the (b)float16 precision.
> Depending on the attention implementation and the precision used, doing a forward pass with a batched request of N tokens (what is done during verification of SD methods) and doing N forward pass of 1 token (vanilla decoding) can lead to very tiny differences in logits.

---

> > ### Comment · Reviewer_Sr5b · 2024-11-25
> >
> > Thanks for the rebuttal. The reviewer have addressed most of the concerns and answered the questions. I will keep the score. Good luck to the authors.

---

> > > ### Author Response · Authors · 2024-11-25
> > >
> > > Dear Reviewer Sr5b,
> > >
> > > We appreciate your approval of our work. Thank you!

---

### Official Review · Reviewer_Wo7q · 2024-10-31

**Soundness:** 4
**Presentation:** 3
**Contribution:** 3
**Rating:** 8
**Confidence:** 4

**Summary:**

This paper focuses on an LLM acceleration technique called Speculative Decoding, which leverages efficient models (smaller but less capable) to draft future tokens, which are verified by the LLM (more capable but much less efficient) in parallel. In particular, it addresses the limitations of previous methods by proposing a Mixture of Attentions architecture on top of a prior work EAGLE to improve its performance. They demonstrate the effectiveness of their approach in both a single-device setting and a novel client-server setting, achieving speedups and improved accuracy. They also present a framework for using LLMs on edge devices, allowing for offline text generation with minimal dependence on a server.

**Strengths:**

- The work improves above EAGLE-2 and seems to achieve state-of-the-art results.
- The work provides a good background on speculative decoding
- The work proposes an interesting client-server setup that fits well with the speculative decoding technique

**Weaknesses:**

- The work lacks an overall view and clear statements that can improve readability.
    - Method intuition: the method section only lays out the information of each component but does not provide an overall view of the proposed method as well as motivating intuitions for each design. The necessary intuitive descriptions are also not found in the appendix.
    - Experiment result: the work only compares to one prior work, EAGLE-2, as a baseline, but did not provide information on how well EAGLE-2 compares with other prior works.
    - the hyperparameter N is used many times. Giving it a name can help readability.
- This work overlooks overall performance metrics, concentrating instead on metrics specific to the speculative decoding framework, such as acceptance length. This focus may inadvertently encourage adversarial scenarios where the smaller model aims merely to deceive the larger model into accepting its outputs rather than genuinely enhancing result quality.
- The work neglects analysis of the time/computational complexity of its method.
- The ablation study compares different hyperparameters but does not ablate the other components. Thus, the importance and contributions of the designs in Section 3.1 and Section 3.3 is not provided.

**Questions:**

(also see weaknesses)
- The experiment on N is arbitrarily set to 0, 1, 3. Why not continuously evaluate 0,1,2.... to the number of layers in the large model?
- In the client-server mode with N > 0, are the last layers of the large model copied onto the client side?

---

> ### Author Response · Authors · 2024-11-20
>
> Thank you very much for your valuable feedback. We are delighted that you found our background on SD informative and the client-server evaluation interesting and useful. Let us address the areas for improvement that will make the paper submission stronger.
>
> > **Q1: The work lacks an overall view and clear statements that can improve readability. Method intuition: the method section only lays out the information of each component but does not provide an overall view of the proposed method as well as motivating intuitions for each design. The necessary intuitive descriptions are also not found in the appendix.**
>
> We will revise the paper to enhance clarity. Here are some additional intuitions of our proposed components.
> * Partial Observability: EAGLE or Medusa only observe a limited (transformed) subset of the whole dynamical state of the LLM. With Layer Self-Attention we can observe the whole (non transformed) state.
> * On-policyness ensures that the small model is trained under conditions that closely resemble the actual inference process, i.e. the LLM state is not observed at every time-step. However, training fully on-policy is expensive because it requires generation during training. With our Cross-Attention solution, we can train more on-policy without having to generate because we can hide specific keys and values.
> * Target Layer Inference: we can reuse the last N layers of the LLM to simplify the inference of the small model by predicting the input of the Nth layer instead of the final layer.
>
> > **Q2: Experiment result: the work only compares to one prior work, EAGLE-2, as a baseline, but did not provide information on how well EAGLE-2 compares with other prior works.**
>
> EAGLE-2 evaluated a wide range of LLMs from 7B to 70B parameters, *which show predictable and consistent improvements across model scales and types*. They also compare against (and improve upon) other methods such as Medusa, Lookahead and Speculative Sampling, establishing themselves as the leading SD methodology [1][2]. Unfortunately, we do not have the same compute capacity as them thus choose one of EAGLE's SOTA LLMs, Llama3, for a fair comparison. We will update Section 4 appropriately, thank you for pointing this out.
>
> [1] Li, Yuhui et al. “EAGLE-2: Faster Inference of Language Models with Dynamic Draft Trees.” Conference on Empirical Methods in Natural Language Processing (2024). https://arxiv.org/pdf/2406.16858
>
> [2] Xia, Heming et al. “Unlocking Efficiency in Large Language Model Inference: A Comprehensive Survey of Speculative Decoding.” Annual Meeting of the Association for Computational Linguistics (2024).
>
> > **Q3: the hyperparameter N is used many times. Giving it a name can help readability.**
>
> We introduce the concept of "Target Layer Inference" in Section 3.2 hence we could replace "N" with "Target Layer", or just "TL" for short. Thanks for the suggestion.
>
> > **Q4: This work overlooks overall performance metrics, concentrating instead on metrics specific to the speculative decoding framework, such as acceptance length. This focus may inadvertently encourage adversarial scenarios where the smaller model aims merely to deceive the larger model into accepting its outputs rather than genuinely enhancing result quality.**
>
> Although it is technically possible that the small model generations could be better than what the large model would generate (in terms of accuracy, safety or other non-speculative metrics), being better than the large model is usually not within the scope of speculative sampling because the small models are assumed to be less capable.
>
> (To be continued ...)

---

> > ### Author Response · Authors · 2024-11-20
> >
> > (Continue from part one...)
> >
> > > **Q5: The work neglects analysis of the time/computational complexity of its method.**
> >
> > Thank you for raising this point.
> > This is usually not studied in SD papers as SD relies on the fact that GPUs are capable of high parallelism (notably during for the verification phase) so even if their theoretical complexity is the same, they can still be faster.
> > Let us analyze the standard decoder-only transformers doing vanilla decoding:
> > * in the first prefill stage, it grows in $\mathcal{O}(L K E (E + K))$ given we have $L$ self-attention layers with $K$ input tokens and an embedding size of $E$
> > * for the K' new decoded tokens, it grows in $\mathcal{O}(\sum_i^{K'} L ( E^2 + E (K + i) ) ) = \mathcal{O}(LE (EK' + K K' + K'^2 )$.
> >
> > If we assume $E$ and $L$ are fixed, it grows in $\mathcal{O}((K+K')^2)$ overall.
> >
> >
> > For speculative decoding, the first prefill stage is the same. Assuming $S$ tokens are verified at a time, the verification would grow in $\mathcal{O}(\sum_i^{\frac{K'}{S}} L ( SE^2 + SE (K + i) ) ) = \mathcal{O}(LE (EK' + K K' + K'^2 )$, leading to the same complexity as vanilla decoding.
> >
> > It dominates the complexity of self-drafting, but we can still analyse it:
> > * For EAGLE, decoding a new token grows in $\mathcal{O}(E^2 + EK)$ as it is a single self-attention layer.
> > * For our Mixture of Attentions architecture, the Self-Attention and Cross-Attention layers also grow in $\mathcal{O}(E^2 + EK)$.
> > The Layer-Self Attention is only called once after every verification stage, so not at every decoding step, it grows in $\mathcal{O}(ALE_{kv}^2+ AE_{kv}L^2)$ if $A$ is the number of accepted tokens in the previous phase.
> >
> > In our experiments, if we look at the first term, $ALE_{kv}^2$ is smaller than $number\\\_of\\\_decoded\\\_tokens \times E^2$ as $E_{kv}$ is 4 times smaller than $E$, $L$ is 32, $A$ is in average 4.5 and $number\\\_of\\\_decoded\\\_tokens$ is 48.
> > Similarly for the second term, $AE_{kv}L^2$ is usually smaller than $number\\\_of\\\_decoded\\\_tokens \times EK$ as soon as the request contains more than $24$ tokens.
> > Therefore, the time complexity is nearly the same as EAGLE overall.
> > We will add this small analysis in the appendix.
> >
> >
> > > **Q6: The ablation study compares different hyperparameters but does not ablate the other components. Thus, the importance and contributions of the designs in Section 3.1 and Section 3.3 is not provided.**
> >
> > Sorry if this was not clear, but we do ablate the solutions proposed in Section 3.1 (Section 3.3 only describes the loss used).
> > In Table 5, you can see the difference in performance between Ours (N=0, -LSA) and EAGLE to understand the importance of training more on-policy with the Cross-Attention Layer (lines 432-435).
> > It provides a major improvement of 26\% in tokens-per-second as well as improved acceptance length of 33\%.
> > Similarly, lines 437-440 focus on checking that Layer Self-Attention is important.
> >
> > > **Q7: The experiment on N is arbitrarily set to 0, 1, 3. Why not continuously evaluate 0,1,2.... to the number of layers in the large model?**
> >
> > Thank you for pointing this out. We will add the N=2 experiment to Table 3 to complete the [0, 1, 2, 3] sequence. In addition, we will aim to add N = [4, 5] to show that going beyond N=3, it is not worth increasing N further due to the trade-off between prediction accuracy and the compute overhead (from running the forward on the last N layers) that we observed in early experiments. Note that N=3 already shows a lack of further benefit. In the single device scenario, the improvement in acceptance length do not compensate the additional time spent on the last 3 layers. In the client-server scenario, the improvement in acceptance length do not compensate for the increase in message length (the KV cache of 3 layers needs to be exchanged).
> >
> > > **Q8: In the client-server mode with N $>$ 0, are the last layers of the large model copied onto the client side?**
> >
> > Yes, the last $N$ layers of the large model are copied to the client.

---

> ### Comment · Reviewer_Wo7q · 2024-11-21
>
> - Q1. Besides intuition of each part, I wonder if the author can provide an overall review of each component, or if they are implemented as a bag of tricks
> - Q2. Since the work only compare with EAGLE, it is important to establish the single reference point (with clear arguments) not only against leading SD methods but also other alternatives (such as lora, quantization, or simply training smaller lm better)
> - Q4: with same concern as Q2, I agree with Reviewer Sr5b that general performance metrics such as accuracy or users study should be included to assess the validity of the final llm output. Only providing SD metrics  leads to the concern that the improvement is only made by fooling the validating model instead of genuine improvement.

---

> > ### Author Response · Authors · 2024-11-22
> >
> > > **Q9. Besides intuition of each part, I wonder if the author can provide an overall review of each component, or if they are implemented as a bag of tricks**
> >
> > The overall view is that we adopt a dynamical system and control perspective to study self-drafting speculative decoding methods.
> > We aim at answering the following questions: what should be the minimal input of a the small model for optimal predictions (we propose Layer Self Attention on the KV cache of the big model), what should be the produced outputs (we propose Target Layer Inference) and how should it be trained optimally (on-policy so we propose Cross-Attention to make it less costly).
> >
> > > **Q10. Since the work only compare with EAGLE, it is important to establish the single reference point (with clear arguments) not only against leading SD methods but also other alternatives (such as lora, quantization, or simply training smaller lm better)**
> >
> > A detailed survey of speculative decoding is beyond the scope of this work. Typically, separate studies present a wider comparison of SD methods, please see [1] and [2] as prime examples.
> >
> > Could you please elaborate how could LORA be an alternative?
> >
> > Speculative decoding methods do not usually compare to quantization methods because we can maintain the same output distribution without quality loss. Moreover, it is an orthogonal approach and could be combined with speculative decoding.
> >
> > We already compare with small independent language models of the same size of our self-drafting small model and trained on the same dataset for the same amount of epoch.
> > It is absolutely possible to train both of those further on more data, but why would we except a flip in prediction performances?
> > The main idea of self-drafting methods is that the input of those small models is already preprocessed by the large model, therefore easing the predictions of the small model.
> > It was already observed by the community and we only reconfirm it here.
> >
> > [1] Xia, Heming, et al. "Unlocking efficiency in large language model inference: A comprehensive survey of speculative decoding." arXiv preprint arXiv:2401.07851 (2024).
> >
> > [2] Zhang, Chen, Zhuorui Liu, and Dawei Song. "Beyond the Speculative Game: A Survey of Speculative Execution in Large Language Models." arXiv preprint arXiv:2404.14897 (2024).
> >
> > > **Q11. with same concern as Q2, I agree with Reviewer Sr5b that general performance metrics such as accuracy or users study should be included to assess the validity of the final llm output. Only providing SD metrics leads to the concern that the improvement is only made by fooling the validating model instead of genuine improvement.**
> >
> > Please see our shared answer https://openreview.net/forum?id=Rz0kozh3LE&noteId=zP4ldvPaIV where we run additional experiments.

---

> > > ### Comment · Reviewer_Wo7q · 2024-11-22
> > >
> > > Thanks. The authors have mostly clarified my concerns. Additional statements like the above which point out that SD maintains output quality while Lora degrades can help the readers quickly grasp an overview of the contribution and position of this work. I have raised my score.

---

> > > > ### Author Response · Authors · 2024-11-24
> > > >
> > > > Thank you for your reply and all the feedback! We sincerely appreciate your approval of our work.

---

### Official Review · Reviewer_2FbK · 2024-11-04

**Soundness:** 3
**Presentation:** 3
**Contribution:** 2
**Rating:** 8
**Confidence:** 2

**Summary:**

This paper proposes a method, Mixture of Attentions for SD, which improves upon the standard SD approach that is used to increase LLM efficiency. The improvements are specifically targeting the partial observability and lack of on-policyness problems of traditional SD. The Mixture of Attentions method is shown to lead to improvements for a single device, but the paper also shows its efficiency/accuracy benefits to a client-server setting.

**Strengths:**

The organization and flow of the paper is very good. The background section is particularly thorough and helpful.

The problem is well-explained (e.g. partial observability and lack of on-policyness are both detailed when explaining the methodology) so it is made clear what exactly the Mixture of Attentions method is aiming to solve. Additionally, the related work is well-addressed. It is clear exactly how this work is different from prior solutions.

It is great that the client-server scenario is tested in a practical setting with different devices (having different resource capacities) and the devices have distance between them. This setting is not only realistic, but it is also well-explained in the text.

There is good theoretical support in the Methodology section, which provides additional justification for the proposed method being superior to the current SOTA (EAGLE, Medusa).

The thorough experiment detail (particularly in the appendix) makes the method highly reproducible.

**Weaknesses:**

The experimentation is very narrow, especially since it only focuses on one model architecture and the improvements over EAGLE seem relatively small and inconsistent. It is therefore not convincing that this method would be effective more generally.

It is not very clear why this problem/contribution is important. The paper would be stronger if the method was motivated by some real-world example where SD may be used, but would lead to significant problems that Mixture of Attentions would mitigate. I understand that the computational requirements of LLMs is an issue, but the introduction could do a better job of explaining why SD should be focused on as a solution for the computational expense and therefore is important to build improvements for. It is also difficult to understand how this work can be have a broader impact or inspire future work. The future work that is suggested at the end of the conclusion seems very specific and narrow. Essentially, the contribution just seems very narrow.

**Questions:**

Why did you choose to only experiment with LLama3-8B-Instruct? It is good that there is justification for only comparing to EAGLE (and not Medusa or Tandem Transformers), but there is no justification for your LLama model choice. How do you think your method would work with other architectures?

---

> ### Author Response · Authors · 2024-11-20
>
> Thank you so much for your valuable feedback. We are really happy that you appreciated the clarity and motivation of the paper as well as our additional client-server experiments aimed to evaluate the method in a very realistic use case. Your enthusiasm for our work is really great so let us address your suggestions for further improvement.
>
> > **Q1: The experimentation is very narrow, especially since it only focuses on one model architecture and the improvements over EAGLE seem relatively small and inconsistent. It is therefore not convincing that this method would be effective more generally.**
>
> In EAGLE [1] and Medusa [2], the authors evaluate a wide range of LLMs from 7B to 70B parameters, **which show predictable and consistent improvements across model scales and types**. Unfortunately, we do not have the same compute capacity as them thus choose one of EAGLE-2's SOTA LLMs, Llama3, for a fair comparison. Nevertheless, we are currently training small models for Qwen2.5-3B-Instruct to confirm that we will obtain similar results. We hope we can obtain the results before the end of the discussion period.
>
> We note that our model is always better than EAGLE-2 for every dataset (with N=1).
> We would also like to emphasise that the speed up ratios are *highly dependent on the hardware used* (notably the CPU because of the complexity of the EAGLE-2 tree-decoding strategy).
> For instance, it was impossible for us to reproduce the 3.46x speedup claimed by EAGLE-2 on our hardware (we achieved 2.14x for the exact same setup).
> This has been reported by others (see https://github.com/SafeAILab/EAGLE/issues/121).
> Therefore, we believe that on more recent CPU, the gap with EAGLE-2 and our architecture would be more significant as we have much better acceptance length.
>
> [1] EAGLE: Speculative Sampling Requires Rethinking Feature Uncertainty
> Yuhui Li, Fangyun Wei, Chao Zhang, Hongyang Zhang.
> International Conference on Machine Learning, 2024.
>
> [2] Medusa: Simple LLM Inference Acceleration Framework with Multiple Decoding Heads
> Tianle Cai, Yuhong Li, Zhengyang Geng, Hongwu Peng, Jason D. Lee, Deming Chen, Tri Dao. International Conference on Machine Learning, 2024.
>
> > **Q2: It is not very clear why this problem/contribution is important. The paper would be stronger if the method was motivated by some real-world example where SD may be used, but would lead to significant problems that Mixture of Attentions would mitigate. I understand that the computational requirements of LLMs is an issue, but the introduction could do a better job of explaining why SD should be focused on as a solution for the computational expense and therefore is important to build improvements for. It is also difficult to understand how this work can be have a broader impact or inspire future work. [...]**
>
> Apologies for any misunderstanding. We believe the client-server evaluation is a good example of an application where our method provides a clear benefit. For example, we beat EAGLE-2 on decoding speeds (up to 80\% on 4G and up to 40\% on 5G) while lowering the calls to the Large LLM by around 50\%. The contribution goes beyond efficiency gains, for example, in the event of a total disconnection, our approach maintains high accuracy compared to other SD methods (15\% - 38\% higher pass@1 score on HumanEval) and demonstrates advantages over API calls to LLMs, which would otherwise be unable to continue the generation process.
> One example could be a user working on a laptop with an LLM integrated system or browser on a train where its internet connection is not stable.
> Another example could be LLM-based translation on mobile phones, etc.
> We will update the introduction accordingly to highlight more concrete use cases.
>
> Regarding the scope of the contributions, although we focused on improving the current SOTA method (EAGLE-2), **our observations (partial observability, on-policyness, target inference layer) are true for many self-drafting methods**, for instance, MEDUSA or MLP Speculator (https://arxiv.org/abs/2404.19124) should also benefit from those.
> Therefore, we believe that self-drafting architectures should be aware of our proposals.
>
> > **Q3: Why did you choose to only experiment with LLama3-8B-Instruct? It is good that there is justification for only comparing to EAGLE (and not Medusa or Tandem Transformers), but there is no justification for your LLama model choice. How do you think your method would work with other architectures?**
>
> We chose Llama3 (8B) as it is a state-of-the-art LLM and because we could compare it to the EAGLE-2 checkpoint. Moreover, there is no fundamental difference between decoder-only transformers architectures that would require significant adaptations to our approach. As previous works showed, the performance of self-drafting methods is consistent across models and model sizes, see answer to Q1. We will update the paper to reflect this.

---

> > ### Author Response · Authors · 2024-11-25
> >
> > Dear Reviewer 2FbKAs,
> >
> > As the rebuttal period comes to an end, we would like to know if our responses have adequately addressed your concerns.
> > We notably run additional experiments with Qwen2.5-3B (please see our shared answer https://openreview.net/forum?id=Rz0kozh3LE&noteId=eamVIL8JnV).
> >
> >
> > Thank you in advance for your time and input.

---

> > > ### Comment · Area_Chair_d18j · 2024-11-26
> > >
> > > Dear reviewer 2FbK,
> > >
> > > Could you please respond to authors' rebuttal and see if you would like to update your review? Thanks very much!
> > >
> > > AC

---

> > > > ### Comment · Reviewer_2FbK · 2024-11-27
> > > >
> > > > Thank you to the authors for taking the time to craft these responses. My questions/concerns have been addressed and I have no further questions. After reading the other reviews and all of the author responses, I decided to raise my rating of the paper.

---

### Official Review · Reviewer_eFLF · 2024-11-08

**Soundness:** 3
**Presentation:** 3
**Contribution:** 3
**Rating:** 6
**Confidence:** 5

**Summary:**

The method addresses two key limitations of existing SD approaches, (1) partial observability and (2) lack of on-policyness, by incorporating Layer Self-Attention (LSA) and Cross-Attention (CA).

**Strengths:**

- The client-server framework with the ability to handle disconnections positions the approach as a practical advancement for deploying LLMs.

- The introduction of LSA and CA layers to mitigate partial observability and improve on-policyness makes sense.

**Weaknesses:**

1. The paper does not thoroughly justify the choice of parameter configurations and its training in its experiments.  As discussed in the Yi et al. (2024), the training dataset and the choice of number of parameters can significantly affect the SD performance, but this paper does not [A].

[A] Yi et al., 2024. Towards Fast Multilingual LLM Inference: Speculative Decoding and Specialized Drafters, EMNLP 2024-main.

2. Discussions for the memory-bound nature of LLM is required in the paper.

3. The effectiveness of the proposed method on models having 3B~13B parameters is unclear. Current experiments focus on relatively smaller models, and the results may not hold for state-of-the-art LLMs, which typically exhibit different scaling dynamics and memory behavior.

4. Typo? line 466.

5. Parallel to Medusa, actually there are concurrent works regrading non-autoregressive heads for SD. It would be good to put discussions for those areas.

- Gloeckle et al. (2024), Better & Faster Large Language Models via Multi-token Prediction.

- Stern et al. (2024), Blockwise Parallel Decoding for Deep Autoregressive Models

- Kim et al. (2024), Accelerating Blockwise Parallel Language Models with Draft Refinement. (https://openreview.net/forum?id=KT6F5Sw0eg)

**Questions:**

See Weakness.

Will update the score after looking at the results of Weakness.

---

> ### Author Response · Authors · 2024-11-20
>
> Thank you for your thorough review and constructive feedback. We appreciate your insights and will address each of the points raised to improve the quality of our paper.
> We are pleased that you acknowledge the importance of our contributions, particularly the client-server framework and the introduction of Layer Self-Attention (LSA) and Cross-Attention (CA) layers to address partial observability and on-policyness.
> We believe those are important for the future development of self-drafting small models.
>
>
> > **Q1. The paper does not thoroughly justify the choice of parameter configurations and its training in its experiments.  As discussed in the Yi et al. (2024), the training dataset and the choice of number of parameters can significantly affect the SD performance.**
>
> We agree that the number of parameters and the training data are crucial.
> That is exactly why we ran our benchmarking in *a very controlled setting*.
> We have trained the relevant baselines (EAGLE-2, Independent small models) and our proposed solutions *using the same training dataset* (Ultrachat 200k) and number of epochs for a fair comparison.
> For reference, we have included the performance of the official EAGLE-2 checkpoint that was trained on the ShareGPT data.
> We did not use ShareGPT as it was unclear which version of the dataset was used.
>
> We also took great care to match the number of model parameters, i.e. "Ours (N=0)", "EAGLE-2", "EAGLE-2 off." and "Independent 1.3B" all have 1.3B parameters (250M trainable and 1.05B frozen, for the "LM head" and "token embed" layers).
> We chose 250M trainable parameters to be directly comparable to EAGLE-2 and their official checkpoint.
> Since increasing N also increases the total number of parameters (frozen in this case), we added a larger "Independent 1.7B" model to compare it with "Ours (N=1)" using 1.55B parameters and "Ours (N=3)" with 1.8B parameters.
> Table 2 clearly shows an advantage for us in both cases.
>
> Finally, *all decoding methods use the same tree-decoding strategy*.
> We will clarify this in the updated version of the paper and include the reference to Yi et al. (2024).
>
> > **Q2. Discussion for the memory-bound nature of LLM is required in the paper.**
>
> Indeed, we will add a paragraph in the background section to explain that for smaller batch sizes, the LLM inference is memory-bound and that speculative decoding can better leverage the spare compute especially with tree-attention.
>
> > **Q3. The effectiveness of the proposed method on models having 3B-13B parameters is unclear. Current experiments focus on relatively smaller models, and the results may not hold for state-of-the-art LLMs, which typically exhibit different scaling dynamics and memory behavior.**
>
> Apologies for any misunderstanding, however, Llama3 (8B) is a SOTA LLM between 3B and 13B. In EAGLE [1] and Medusa [2], the authors evaluate a wide range of LLMs from 7B to 70B parameters, **which show predictable and consistent improvements across model scales and types**. Unfortunately, we do not have the same compute capacity as them thus choose one of EAGLE-2's SOTA LLMs, Llama3, for a fair comparison. Moreover, our small models are bounded by the practicalities of small device memory for the client-server setting. Nevertheless, we are currently training small models for Qwen2.5-3B-Instruct to confirm that we will obtain similar results. We hope we can obtain the results before the end of the discussion period.
>
> [1] EAGLE: Speculative Sampling Requires Rethinking Feature Uncertainty
> Yuhui Li, Fangyun Wei, Chao Zhang, Hongyang Zhang.
> International Conference on Machine Learning, 2024.
>
> [2] Medusa: Simple LLM Inference Acceleration Framework with Multiple Decoding Heads
> Tianle Cai, Yuhong Li, Zhengyang Geng, Hongwu Peng, Jason D. Lee, Deming Chen, Tri Dao. International Conference on Machine Learning, 2024.
>
> > **Q4. Typo? line 466.**
>
> Sorry, could you please quote the exact string to help us pinpoint the possible typo? Thank you.
>
> > **Q5: Parallel to Medusa, actually there are concurrent works regrading non-autoregressive heads for SD. It would be good to put discussions for those areas.**
>
> Thank you so much for the citations. Rest assured we will add them to the discussions in related work.
>
> We believe our proposed Layer Self Attention and Target Layer Inference could to easily be applied in [1].
> Similarly, we could also mitigate partial observability in [2] with Layer Self Attention.
>
> [1] Gloeckle et al. (2024), Better \& Faster Large Language Models via Multi-token Prediction.
>
> [2] Kim et al. (2024), Accelerating Blockwise Parallel Language Models with Draft Refinement.

---

> ### Comment · Reviewer_eFLF · 2024-11-25
>
> Thank you for your thorough responses to the points I raised. Upon reviewing your clarifications and additional explanations, I realize that some of my earlier concerns were based on misunderstandings or insufficiently detailed observations. I appreciate the effort you have made to provide clear and specific information to address the weaknesses I highlighted. The practical advancements in client-server frameworks and the methodological contributions addressing partial observability and on-policyness are significant and relevant to the field. I raised the score, accordingly.

---

> > ### Author Response · Authors · 2024-11-25
> >
> > Dear Reviewer eFLF,
> >
> > Thank you for taking the time to review our paper and for your approval of our work.

---

### Author Response · Authors · 2024-11-21

We would like to answer a shared concern about the possible loss in generation quality with speculative decoding methods (reviewer Wo7q.Q4 and Sr5b.Q5). To do so, we ran several experiments to assess the quality of the generated responses using greedy decoding. We focused on 3 datasets from SpecBench (HumanEval, GSM8K and CNN/DM) that do not require access to proprietary models/APIs for evaluation (llm-as-a-judge).


| Vanilla decoding | HumanEval (pass@1) | GSM8K (accuracy) | CNN/DM (Rouge-L f-score) |
| ----------- | :----: | :----: | :----: |
| Llama-8B | 62.5% | 80% | 0.3071 |

| Speculative Decoding | HumanEval (pass@1) | GSM8K (accuracy) | CNN/DM (Rouge-L f-score) |
| ----------- | :----: | :----: | :----: |
| Ours (N=3) | 62.5% | 80% | 0.3053 |
| Ours (N=1)   | 62.5% | 81.25% | 0.3068 |
| Ours (N=0)   | 62.5%  | 80% | 0.3070 |
| EAGLE-2   | 62.5%  | 81.25% | 0.3062 |
| EAGLE-2 off   | 62.5%  | 80% | 0.3056 |
| Independent 1.7B   | 62.5%  | 80% | 0.3067 |
| Independent 1.3B   | 62.5%  | 80% | 0.3064 |

As you can observe in the tables above, the pass@1 on HumanEval is the same across all methods. The accuracy on GSM8K actually improves w.r.t the base model on one question for Ours (N=1) and EAGLE-2. Finally, the ROUGE scores are also extremely similar, leading us to conclude that any differences to the base model are negligible and almost certainly appear due to using float16 (see our answer to Sr5b.Q5 for more details).

---

### Author Response · Authors · 2024-11-25
**Qwen2.5-3B**

Dear reviewers,

Thank you very much for your suggestions to make the paper stronger.
We would like to answer a shared concern about the limited range of models and sizes that we tested (reviewer eFLF.Q3 and 2FbK.Q3).
We trained 4 additional small models on the Ultrachat dataset to accelerate Qwen2.5 3B.
EAGLE recommends to use one decoder layer of the big LLM to define the size of the small LM, which leads to a trainable size of 80M parameters.
We kept the shared "embed_tokens/LM_head" layer frozen.

| Small Model | Total size | Trainable size   | MT-bench    |     | HumanEval |          | GSM8K    |          | Alpaca   |          | CNN/DM   |          | Natural Ques. |          | Mean     |          |
|--------------|-------|-----------|----------|----------|-----------|----------|----------|----------|----------|----------|----------|----------|---------------|----------|----------|----------|
| | | | Speedup  | $\tau$   | Speedup   | $\tau$   | Speedup  | $\tau$   | Speedup  | $\tau$   | Speedup  | $\tau$   | Speedup       | $\tau$   | Speedup  | $\tau$   |
| Ours (N=1)   | 0.48B  | 80M      | 1.7     | **3.61**     | 2.08    | **4.5**     | 1.5     | **3.31**     | 1.89 | **4.02**     | 1.78 | **3.81**     | 1.71      | **3.7**     | 1.76 | **3.82**     |
| Ours (N=0)   | 0.4B  | 80M      | **1.77**     | 3.56     | **2.16**    | 4.45     | **1.58**     | 3.24     | **1.89** | 3.82     | **1.83** | 3.71     | **1.69**      | 3.48     | **1.82** | 3.70     |
| EAGLE-2      | 0.4B  | 80M      | 1.65     | 3.13     | 1.88      | 3.57     | 1.5     | 3    | 1.78     | 3.35     | 1.7     | 3.14     | 1.63          | 3.05     | 1.69     | 3.2     |
| Independent  | 0.4B  | 80M      | 1.49     | 2.9     | 1.8      | 3.6     | 1.29     | 2.56     | 1.66     | 3.22     | 1.46     | 2.84     | 1.51          | 2.99     | 1.53     | 3.01     |

We can observe that in this setting, "Ours N=0" gives the highest speedup while "Ours N=1" has better acceptance length.
The trend we observed on Llama3-8B-Instruct is **maintained** (we are better than "EAGLE-2" and "EAGLE-2" is better than "Independent").
Once again, we would like to highlight that the speedup is highly dependent on the hardware used and we believe the gap will be bigger on more recent CPUs (see answer 2FbK.Q1 for more details https://openreview.net/forum?id=Rz0kozh3LE&noteId=yGJpPvjANr).

---

### Public Comment · ~Penghui_Yang1 · 2025-02-05
**Clarification Needed on Overlap with ICML 2024 Work**

Thank you for presenting this work. Upon reviewing the methodology described, I noticed striking similarities with the ICML 2024 paper, *GliDe with a CaPE: A Low-Hassle Method to Accelerate Speculative Decoding*. Both works appear to share a key process: draft tokens are processed through the large model's embedding layer, followed by self-attention to generate queries, which are then used to compute cross-attention with key-value pairs obtained from the large model.

Given the strong resemblance, I believe it would greatly benefit the scientific discussion if this paper explicitly addressed the ICML 2024 work. A detailed comparison between the two would clarify the relationship and allow readers to better understand the unique contributions of this paper. Specifically, it would be helpful to highlight the novel aspects introduced here that are distinct from those in *GliDe with a CaPE*.

Transparency and proper contextualization are critical for advancing the field, and I believe addressing this will further strengthen the paper. I look forward to seeing this clarification.

---

> ### Public Comment · ~Matthieu_Zimmer1 · 2025-02-28
>
> Thank you for sharing this paper; we were not aware of it.
> We have added it to the related work section and discussed the differences with it.
> We aim to include a comparison with this approach, unfortunately no trained checkpoints are available online. It will take some time to train it in the same scenario for a fair comparison.

---

> > ### Public Comment · ~Penghui_Yang1 · 2025-03-01
> >
> > Thank you for your response and for adding the citation. However, I have noticed a new issue regarding the way GliDe’s methodology is described in your updated discussion. I will detail this concern in a separate public comment for better visibility. Looking forward to your clarification.

---

### Public Comment · ~Penghui_Yang1 · 2025-03-01
**Misrepresentation of GliDe’s Methodology in the Camera-Ready Version**

I would like to clarify a factual inaccuracy in the newly added comparison.

In the updated version, the authors claim that *Du et al. (2024)* (i.e., GliDe) leverages the KV-cache of all layers of $M_{\text{Large}}$, implying that it requires as many layers in $M_{\text{Small}}$ as in $M_{\text{Large}}$ to fully capture the state. **This is incorrect.** GliDe explicitly states that it **only uses the KV-cache from the final layer of $M_{\text{Large}}$**, not all layers, and GliDe conducts an ablation study about this in Figure 7.

This misrepresentation significantly affects the validity of the comparison. The argument that GliDe would result in slow drafting due to needing all layers is based on a false premise. I strongly encourage the authors to **correct this misunderstanding and provide a more accurate discussion** of GliDe’s approach.

While the authors mention plans for an empirical comparison, at this stage, the two approaches appear **highly similar**, and I find it difficult to identify clear distinctions between them. I would appreciate it if the authors could further elaborate on the key differences to provide a clearer understanding.

Looking forward to the authors’ clarification.

---

> ### Public Comment · ~Matthieu_Zimmer1 · 2025-03-14
>
> Thank you for your message.
>
> We did not write that Glide always needs the same number of layers in $M_\text{Small}$ as what is in $M_\text{Large}$. What we wrote is that, **if Glide wanted to fully capture the state of the dynamical system**, **which you should do if you want to maximize acceptance rate** [_our first contribution is to identify this_], then Glide will need the same number of layers inside $M_\text{Small}$ and $M_\text{Large}$ according to their architecture description.
>
> Now, as in practice they only use the last layer KV-cache, they will suffer from partial observability (see our paper). We confirmed this by adding Glide in our experiments (Table 2). It is slower than EAGLE-2 but has higher acceptance rate. However, compared to us, their acceptance rate is indeed lower as we would expect from our theory.
>
> So, how can we capture the state of the dynamical system without needing many layers in $M_\text{Small}$? With our proposed Layer Self Attention layer [_second contribution/key difference_].
>
> **Therefore, what we wrote in the discussion section is correct.**
>
> Regarding other distinctions with Glide, we also propose target layer inference, and we try to always justify our design choices with formal properties (for instance why is it important to have a self attention layer before the cross attention one).
> We also focused on client-server scenarios and released the source code of the 4 methods (Mixture of Attentions, EAGLE-2, Glide and Independent) within a single framework.

---

### Meta-Review · Area_Chair_d18j · 2024-12-23

**Metareview:**

All reviewers agreed that this is an good contribution to speculative decoding.
Strength:
1. An in-depth analysis of why new methodologies could be needed to improve upon the existing SD solutions.
2. An novel solution to mitigate the issues discussed.

Weakness:
1. Improvement over SOTA is not big. Given that the methodology is much more complex, so the potential impact can be small.

**Additional Comments On Reviewer Discussion:**

All reviewers actively participated the discussions and concerns were mostly addressed by the rebuttal.

---

### Decision · Program_Chairs · 2025-01-22

Accept (Poster)